# Mitigating Noise-Induced Layout Priors for Object Counting in Diffusion Models

Xiaoling Gu [* 1]   Xuelong Li [* 1]   Shengqi Wu [1]
Yongkang Wong [2]   Zizhao Wu [3]   Huan Li [4]   Zhou Yu [1]   Mohan Kankanhalli [2]

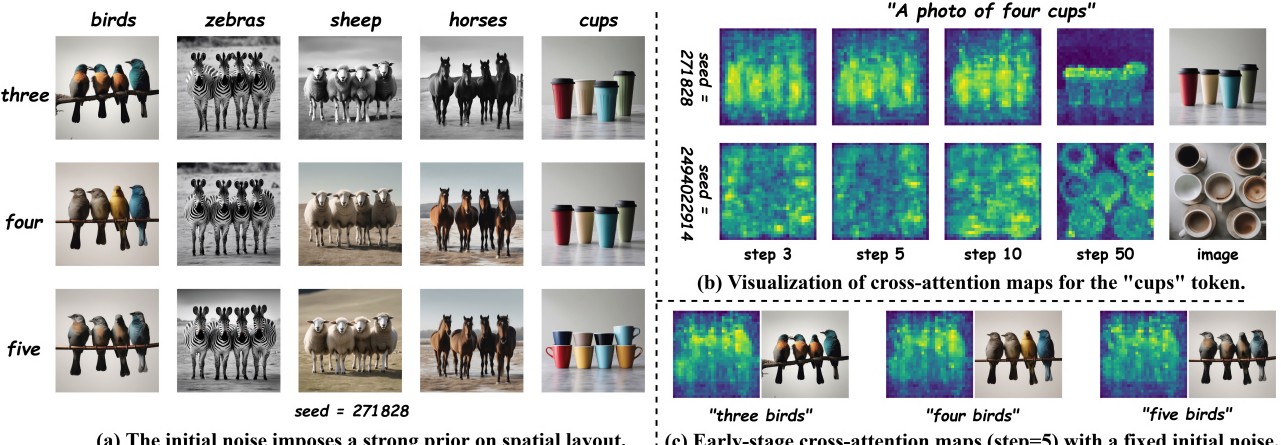

Figure 1. Initial noise strongly governs spatial layout and object count in text-to-image diffusion models. (a) With fixed initial noise, changing the object count in the prompt produces nearly identical layouts. (b) Early-stage cross-attention maps with well-separated, count-aligned responses facilitate correct object counting, while scattered responses tend to result in counting errors. (c) When the initial noise is fixed, early cross-attention maps remain highly similar despite different object-count prompts, revealing a noise-conditioned attention pattern that constrains layout formation.

## Abstract

Despite remarkable progress in text-to-image diffusion models, accurately generating the specified number of objects remains a persistent challenge. We identify the initial noise as a primary determinant of spatial layout formation, with early-stage cross-attention serving as the key mechanism that mediates the propagation of noise-induced structures throughout the denoising process. We characterize this phenomenon as *Noise-Induced Layout Prior*. Leveraging this insight, we propose a novel training-free framework for object counting in diffusion models. Our approach consists of two key components: (1) a *Count-Aware*

*Noise Adjustment Strategy*, which explicitly manipulates the initial latent noise to align layout formation with the target object count, and (2) an *Attention-Guided Layout Consistency Strategy*, which performs test-time optimization on early-stage cross-attention to further stabilize layout formation during denoising. Extensive experiments on both single-category and multi-category benchmarks demonstrate that our method consistently outperforms strong diffusion baselines and state-of-the-art object count control methods in terms of counting accuracy and image quality. Code Release: https://github.com/lxlong1201/Mitigate_Noise_Prior.

*Equal contribution  [1]Zhejiang Key Laboratory of Space Information Sensing and Transmission, Hangzhou Dianzi University, Hangzhou, 310018, China [2]National University of Singapore, Singapore [3]School of Digital Media Technology, Hangzhou Dianzi University, Hangzhou, China [4]Zhejiang University, Hangzhou, China. Correspondence to: Zhou Yu <yuz@hdu.edu.cn>.

*Proceedings of the 43rd International Conference on Machine Learning*, Seoul, South Korea. PMLR 306, 2026. Copyright 2026 by the author(s).

## 1. Introduction

Text-to-image (T2I) diffusion models have achieved significant progress in generating high-fidelity and diverse images from natural language prompts (Nichol & Dhariwal, 2021; Rombach et al., 2022; Ramesh et al., 2022; Saharia et al., 2022; Nichol et al., 2022). Despite these advances, they consistently struggle to generate the precise number

of objects specified in a prompt (Dahary et al., 2024; Li et al., 2024a). This limitation is pervasive across model families, affecting recent state-of-the-art models including SDXL (Podell et al., 2024), Stable Diffusion 3.0 (Esser et al., 2024) and FLUX (Labs, 2024). Notably, this failure arises even in seemingly trivial cases: prompts like "three birds" frequently result in images containing an incorrect number of objects (see Figure 1).

To address object counting errors, prior work has explored two strategies. One line of research adopts a two-stage text-to-layout and layout-to-image pipeline (Phung et al., 2024; Binyamin et al., 2025), explicitly inferring or refining object layouts from the prompt. The other approaches apply direct guidance during diffusion, such as Counting Guidance (Kang et al., 2025), which estimates object counts at each denoising step and adjusts the generation trajectory accordingly. Despite their differences, these methods largely assume that counting errors can be mitigated via layout refinement or conditional guidance during denoising.

However, we identify a critical yet underexplored factor driving object counting errors: the initial noise exerts a dominant influence on the spatial layout and object count. As shown in Figure 1(a), fixing the initial noise while varying the object count in the prompt (e.g., from "three" to "five") compels models like SDXL to produce nearly identical layouts. This indicates that once a coarse object-level structure is established via the initial noise, subsequent text conditioning struggles to override it, resulting in weak adherence to numerical constraints.

To further investigate this prior effect, we analyze cross-attention maps at different denoising stages under a fixed prompt while varying the random seed. As shown in Figure 1(b), early-stage cross-attention maps that exhibit well-separated and count-aligned high-response regions are strongly associated with accurate object counts. In contrast, scattered or misaligned attention patterns frequently correspond to failures in satisfying the numerical constraints specified in the prompt. These observations are consistent with prior studies (Kim et al., 2023; Battash et al., 2024), which show that T2I diffusion models establish coarse object layouts early in denoising and refine visual details in later stages. Consequently, effective object counting requires intervention at the earliest stages of generation, before object layouts become fixed.

Beyond identifying the noise-to-layout correlation, we probe the underlying mechanism governing this effect. In particular, we examine how early-stage cross-attention responses form under controlled noise conditions. As shown in Figure 1(c), fixing the initial noise yields strikingly persistent early-stage cross-attention maps despite variations in the object count specified in the prompt. This reveals that early-stage patterns are not determined solely by the prompt,

but are tightly coupled with the sampled noise. These observations suggest that early-stage cross-attention acts as a primary mediator, propagating noise-induced structures throughout the denoising process. We term this phenomenon **Noise-Induced Layout Prior**.

Motivated by these observations, we propose a novel training-free framework for object counting in text-to-image diffusion models, which intervenes in object arrangement during the early stages of denoising. Specifically, we design a **Count-Aware Noise Adjustment Strategy** that aligns object layouts with the target object count by manipulating the initial noise. This strategy begins with an *object region planning* module that partitions the latent space according to the target object count, while preserving the spatial priors embedded in the initial noise. Guided by this partition, we introduce two complementary modules, *latent representation reorganization* and *object saliency enhancement*, which together encourage the emergence of distinct object instances within each designated region. To further promote layout consistency during denoising, we introduce an **Attention-Guided Layout Consistency Strategy** that complements the noise adjustment process by performing test-time optimization on early-stage cross-attention.

Our main contributions are summarized as follows:

- We provide a systematic analysis of the role of initial noise in T2I diffusion models, demonstrating that it strongly governs spatial layout and often dominates object count control during the denoising process.

- We propose a novel training-free framework for object counting in T2I diffusion models. The proposed approach integrates a *count-aware noise adjustment strategy* with an *attention-guided layout consistency strategy*, enabling the denoising process to generate object layouts that align with the target object count specified in the prompt.

- Extensive experiments on both single-category and multi-category object counting benchmarks show that our method consistently outperforms strong diffusion-based baselines and state-of-the-art object count control methods.

## 2. Related Work

### 2.1. Object Count Control in T2I Generation

Recent efforts have sought to improve object counting in diffusion-based T2I models, primarily through explicit structural guidance or test-time interventions. A common approach decomposes generation into text-to-layout and layout-to-image stages, introducing intermediate layout representations as control signals. For example, Attention Re-

focus (Phung et al., 2024) infers object bounding boxes from prompts using large language models and regularizes cross-attention during denoising. CountLoop (Mondal et al., 2025) uses a VLM-guided agent to iteratively refine layouts for high-density object counting. CountDiffusion (Li et al., 2025b) first derives count-inconsistent regions from a clean image and then guides attention-based correction. CountGen (Binyamin et al., 2025) predicts object masks via a learned layout module followed by test-time refinement. Be Decisive (Dahary et al., 2025) derives layouts from noisy latent to guide multi-subject generation. CountCluster (Lee et al., 2025) clusters cross-attention maps into count-aware layouts to constrain first-step attention distributions. Alternatively, direct guidance methods operate during denoising, such as Counting Guidance (Kang et al., 2025), which uses a pre-trained counting model for gradient-based control, and IoCo (Zafar et al., 2026), which iteratively optimizes count-related prompt tokens. In contrast, our method introduces a novel training-free framework that intervenes in object layout during the early stages of denoising, enabling precise object arrangement without auxiliary models or layout supervision.

## 2.2. Initial Noise in T2I Generation

Recent studies have highlighted the critical role of initial noise in diffusion-based T2I generation. GoodSeed (Xu et al., 2025) demonstrates that different random seeds can lead to large variations in visual quality, while Crystal Ball (Ban et al., 2025) and subsequent work (Mao et al., 2024) show that localized *trigger patches* in the initial noise can deterministically induce object generation. Motivated by these findings, several methods optimize or select initial noise to improve controllability, including gradient-based noise optimization (Eyring et al., 2024), attention-driven noise partitioning (Guo et al., 2024), and reliable seed mining (Li et al., 2025a). However, these approaches primarily target visual quality or text–image alignment, and do not explicitly model object count, limiting their ability to achieve accurate numerical control.

## 3. Methodology

### 3.1. Preliminaries and Observations

Let $\mathcal{G}$ denote a T2I diffusion model that generates an image $I = \mathcal{G}(z_T, y)$, where $z_T \sim \mathcal{N}(0, \mathbf{I})$ is the initial latent noise and $y$ is the textual prompt containing a target object count $C$. During the denoising process at timestep $t$, the cross-attention map for an object token $w \in y$ is denoted as $\mathbf{A}_t \in \mathbb{R}^{H \times W}$. As analyzed in Section 1, the early denoising stage ($t \approx T$) is the critical period for establishing the coarse spatial layout. We summarize our observations as follows:

***Empirical Observation 1** (Noise-Induced Layout Prior).*

For a fixed initial noise $z_T$, varying the target object count $C$ in prompts $\{y_i, y_j\}$ results in cross-attention maps $\mathbf{A}_t$ that exhibit a persistent and highly similar spatial structure $\mathcal{S}$ during the early denoising stages ($t > T - \Delta t$). This indicates that the initial noise $z_T$ exerts a dominant influence on the spatial layout, creating a *prior* that often overrides the numerical constraints specified in the textual prompt $y$.

### 3.2. Framework Overview

To mitigate the *Noise-Induced Layout Prior*, we propose a novel training-free framework for object counting in diffusion models that explicitly intervenes in object arrangement during the early stages of denoising. As illustrated in Figure 2, our approach introduces a *Count-Aware Noise Adjustment Strategy* that manipulates the initial noise to guide the model toward object layouts consistent with the target object count. To further enhance layout stability throughout denoising, we incorporate an *Attention-Guided Layout Consistency Strategy*, which performs test-time optimization on early-stage cross-attention. These two components jointly regulate early denoising dynamics, leading to substantially improved object counting accuracy.

### 3.3. Count-Aware Noise Adjustment Strategy

In this stage, we first introduce an *Object Region Planning* module that partitions the latent space according to the target object count while preserving the spatial layout embedded in the initial noise. Based on this partition, we further refine the initial noise through two complementary modules: *Latent Representation Reorganization* (LRR) and *Object Saliency Enhancement* (OSE), which together encourage the emergence of distinct object instances within each designated region.

**Object Region Planning.** Given a prompt such as "a photo of three apples on the ground", we run $S$ denoising steps and aggregate the cross-attention maps associated with the object token (e.g., "apple") by averaging them over timesteps, yielding $\bar{A}_S \in \mathbb{R}^{H \times W}$. We then apply min-max normalization to obtain $\hat{A}_S$, which is thresholded to generate a high-response mask:

$$M_{\text{attn}} = \mathbb{I}\left(\hat{A}_S \geq \tau\right), \ \ M_{\text{attn}} \in \{0,1\}^{H \times W}. \quad (1)$$

This mask captures the model's spatial preference for object placement. Here, $\tau$ is a threshold that selects spatial locations with high attention responses, and $\mathbb{I}(\cdot)$ denotes the indicator function.

Next, we apply the K-means clustering algorithm (MacQueen, 1967) to the coordinates within $M_{\text{attn}}$, using $\hat{A}_S$ as attention-weighted distances to guide the clustering. This yields $K$ cluster centers $\{c^{(i)}\}_{i=1}^{K}$ and corresponding cluster masks $\{M_{\text{attn}}^{(i)}\}_{i=1}^{K}$, where $K$ is the target object count.

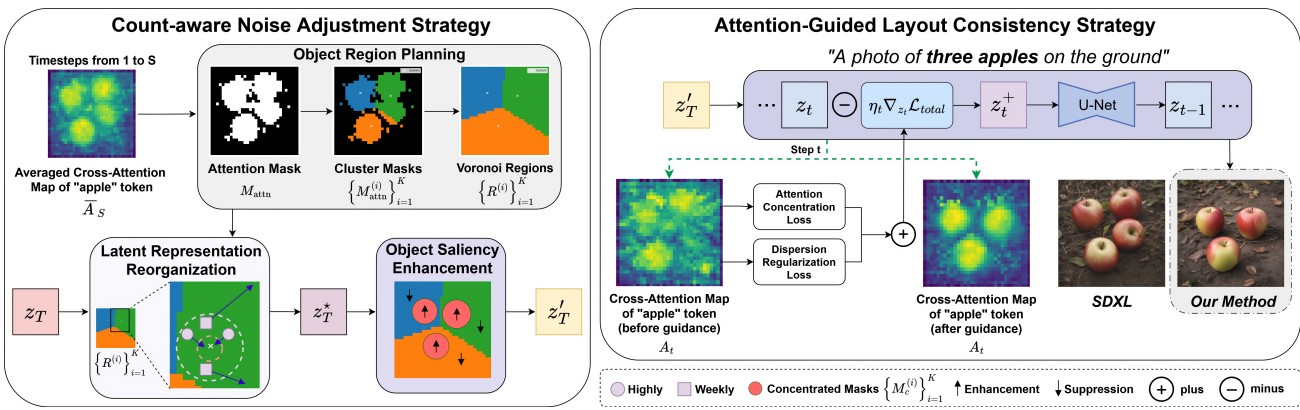

*Figure 2.* **Architecture outline**. Given a prompt specifying an object count, the *Count-Aware Noise Adjustment Strategy* steers object layout formation toward the target object count by manipulating the initial noise. The *Attention-Guided Layout Consistency Strategy* further promotes layout consistency through test-time optimization of early-stage cross-attention.

Based on these centers, we construct a Voronoi diagram (Aurenhammer, 1991) over the latent space, yielding $K$ mutually exclusive regions $\{R^{(i)}\}_{i=1}^K$, each intended to host a distinct object instance.

**Latent Representation Reorganization.** LRR reshapes the initial noise by reorganizing spatial latent features within each object region according to their attention strength. The key intuition is to concentrate highly object-related latent features toward the region center, while pushing less relevant features outward, thereby enhancing intra-object coherence and inter-object separation. Given the early-stage cross-attention map $\bar{A}_S$, we first normalize attention values within each object region $R^{(i)}$ to obtain $\hat{A}_S^{(i)}$. Latent features within $R^{(i)}$ are then reorganized by permuting their spatial assignments based on two criteria: attention magnitude and spatial distance to the region center $c^{(i)}$. Specifically, features with higher attention scores are reassigned to spatial locations closer to $c^{(i)}$, while lower-attention features are placed farther away. Formally, this reorganization is expressed as:

$$z_T^*(p_{[j]}^{(i)}) \leftarrow z_T(q_{[j]}^{(i)}). \qquad (2)$$

where $\{q_{[j]}^{(i)}\}$ and $\{p_{[j]}^{(i)}\}$ denote spatial positions sorted by attention (descending) and distance to the region center (ascending), respectively.

Applying this permutation to all object regions yields the reorganized noise representation $z_T^*$.

**Object Saliency Enhancement.** The core idea of OSE is to further refine the initial noise by amplifying saliency cues in regions likely to host object instances. To coarsely distinguish object instances from background within each region $R^{(i)}$, we define a concentrated mask $M_c^{(i)}$ as a circular area centered at the region center $c^{(i)}$. The radius $r^{(i)}$ is adaptively determined by the spatial extent of the corresponding

attention cluster mask $M_{attn}^{(i)}$, we use $|\cdot|$ to compute the area:

$$M_c^{(i)} = \text{Circle}(c^{(i)}, r^{(i)}), \;\; r^{(i)} = \max\left(1, \sqrt{\frac{|M_{attn}^{(i)}|}{\pi}}\right). \qquad (3)$$

Given the concentrated masks $\{M_c^{(i)}\}_{i=1}^K$, we apply a saliency-aware modulation to the reorganized latent noise $z_T^*$. Specifically, latent features are reweighted based on whether their spatial locations fall within any concentrated mask:

$$z_T'(l) = \begin{cases} \mu + \gamma \cdot (z_T^*(l) - \mu) & \text{if } l \in \bigcup_i M_c^{(i)}, \\ \mu + \beta \cdot (z_T^*(l) - \mu) & \text{otherwise.} \end{cases} \qquad (4)$$

where $\mu$ denotes the mean of latent features, and $\gamma > 1$ and $\beta < 1$ respectively amplify object-relevant regions and mildly suppress background activations. Together with LRR, OSE reinforces object-level saliency while preserving the planned layout, guiding the diffusion process toward an object configuration consistent with the target object count.

### 3.4. Attention-Guided Layout Consistency Strategy

Although the noise adjustment strategy guides object layouts toward the target object count while preserving overall structure, its effectiveness can degrade due to dispersed attention and ambiguous object boundaries. To address this, we introduce *Attention Concentration Loss* and *Dispersion Regularization Loss*, which encourage compact, well-defined attention within each region, promoting the emergence of a single coherent object per region.

Given a cross-attention map $A_t \in \mathbb{R}^{H \times W}$ associated with a specific object at denoising step $t$, we restrict $A_t$ to region $R^{(i)}$ to obtain the region-specific attention map $A_t^{(i)}$. We then apply min-max normalization within the region to obtain the normalized attention map $\hat{A}_t^{(i)}$. Next, Otsu's

method (Otsu, 1979) is applied to $\hat{A}_t^{(i)}$ to automatically determine a threshold $\tau_{\text{otsu}}$ that separates foreground from the background.

To filter weak attention responses and focus on highly activated regions, we introduce a scaling factor $\lambda_o$ and retain only the locations whose attention values exceed the adjusted threshold $\lambda_o \tau_{\text{otsu}}$. Formally, for $(x, y) \in R^{(i)}$:

$$\tilde{A}_t^{(i)}(x, y) = \begin{cases} \hat{A}_t^{(i)}(x, y) & \text{if } \hat{A}_t^{(i)}(x, y) \geq \lambda_o \tau_{\text{otsu}}, \\ 0 & \text{otherwise.} \end{cases} \quad (5)$$

**Attention Concentration Loss.** This loss consists of two components: a positive term $f_{\text{pos}}^{(i)}$ that encourages attention to concentrate around the centroid of region $R^{(i)}$, and a negative term $f_{\text{neg}}^{(i)}$ that penalizes attention drift toward other regions. To encourage the attention distribution to converge toward the predicted object centers, we define a spatial concentration operator $\mathcal{F}$. This operator computes the intensity-weighted variance of the attention map $\tilde{A}_t^{(i)}$ relative to its regional centroid $c_t^{(i)}$:

$$\mathcal{F}\left(\tilde{A}_t^{(i)}, c_t^{(i)}\right) = w_{(i)} \sum_{(x,y) \in R^{(i)}} \frac{\tilde{A}_t^{(i)}(x, y) \left\| (x, y) - c_t^{(i)} \right\|^2}{Z_{(i)}}. \quad (6)$$

where $Z_{(i)} = \sum_{(x,y) \in R^{(i)}} \tilde{A}_t^{(i)}(x, y)$ is the total aggregated attention within region $R^{(i)}$, and $w_{(i)} = \frac{HW}{|R^{(i)}|}$ is a scale-normalization factor.

Using this definition, the positive and negative components are given by:

$$f_{\text{pos}}^{(i)} = \mathcal{F}\left(\tilde{A}_t^{(i)}, c_t^{(i)}\right), \quad f_{\text{neg}}^{(i,j)} = \mathcal{F}\left(\tilde{A}_t^{(i)}, c_t^{(j)}\right). \quad (7)$$

where $j \neq i$ indexes competing regions.

To focus the penalty on the most interfering regions, we aggregate the negative terms using a Softmax-weighted scheme:

$$f_{\text{neg}}^{(i)} = \sum_{\substack{j=1 \\ j \neq i}}^{K} v^{(i,j)} f_{\text{neg}}^{(i,j)}, \quad v^{(i)} = \text{Softmax}(-\lambda_s f_{\text{neg}}^{(i,j)}). \quad (8)$$

where $\lambda_s$ is a temperature hyperparameter that controls the sharpness of the weighting distribution. Finally, the region-wise *Attention Concentration Loss* is defined as:

$$\mathcal{L}_{\text{acl}}^{(i)} = \max\left(f_{\text{pos}}^{(i)} - \lambda_d f_{\text{neg}}^{(i)}, 0\right). \quad (9)$$

where $\lambda_d$ balances the positive and negative components.

**Dispersion Regularization Loss.** This loss complements the *Attention Concentration Loss* by penalizing attention

maps that are spatially over-extended, even when their centroids are localized. We define the dispersion as the ratio between the effective spatial occupancy and the total attention mass:

$$\mathcal{L}_{\text{drl}}^{(i)} = \frac{\left| (x, y) \in R^{(i)} \mid \tilde{A}_t^{(i)}(x, y) > 0 \right|}{Z_{(i)} + \eta}. \quad (10)$$

where the numerator measures the spatial support of the attention map $\tilde{A}_t^{(i)}(x, y)$ within region $R^{(i)}$, i.e., the number of spatial locations receiving non-zero attention. The denominator represents the total aggregated attention within region $R^{(i)}$, and $\eta$ is a small constant for numerical stability.

**Semantic Suppression Loss.** In multi-category object counting, a single region may respond to multiple object tokens, causing object leakage across regions. For instance, in the prompt "three apples and four peaches", a region intended for apples may incorrectly attend to peach-related attributes. To mitigate this issue, we introduce a *Semantic Suppression Loss* that penalizes attention activations from irrelevant object categories within each region.

Let $\phi(i) \in \mathcal{N}$ denote the object category assigned to region $R^{(i)}$ according to the region–category correspondence. For any object category $n \in \mathcal{N}$, we define the average attention response within region $R^{(i)}$ as:

$$S_n^{(i)} = \frac{\sum_{(x,y) \in R^{(i)}} \tilde{A}_{t,n}^{(i)}(x, y)}{\left| (x, y) \in R^{(i)} \mid \tilde{A}_{t,n}^{(i)}(x, y) > 0 \right|}. \quad (11)$$

where $\tilde{A}_{t,n}^{(i)}$ denotes the filtered attention value of object category $n$ within region $R^{(i)}$.

The region-wise *Semantic Suppression Loss* is defined as:

$$\mathcal{L}_{\text{ssl}}^{(i)} = \frac{1}{|\mathcal{N}| - 1} \sum_{\substack{n' \in \mathcal{N} \\ n' \neq \phi(i)}} S_{n'}^{(i)}. \quad (12)$$

where $|\mathcal{N}|$ denotes the number of object categories in the prompt. This loss penalizes attention responses from irrelevant object categories $n' \neq \phi(i)$ within region $R^{(i)}$, encouraging each region to focus exclusively on its designated object category.

**Total Objective Function.** Given a set of region-wise losses $\{\mathcal{L}_{\text{acl}}^{(i)}\}_{i=1}^{K}$, we aggregate them into a global objective using a Softmax-weighted scheme:

$$\mathcal{L}_{\text{acl}} = \sum_{i=1}^{K} w^{(i)} \mathcal{L}_{\text{acl}}^{(i)}, \quad w^{(i)} = \text{Softmax}(\lambda_s \mathcal{L}_{\text{acl}}^{(i)}). \quad (13)$$

Here, $\lambda_s$ is a temperature hyperparameter that controls the sharpness of the weighting distribution. This approach dynamically prioritizes regions with higher error values, ensuring the optimization focuses on the most significant layout

*Table 1.* Quantitative results on the **COCOCount** benchmark.

| Method | STA↑ | Acc↑ | MAE↓ | Precision↑ | Recall↑ | F1↑ | IR↑ |
|---|---|---|---|---|---|---|---|
| FLUX.1 (Labs, 2024) | 0.8069 | 0.4425 | 1.1317 | 0.9151 | 0.9330 | 0.9081 | 1.2027 |
| PixArt-Σ (Chen et al., 2024) | 0.7690 | 0.3950 | 1.4258 | 0.9109 | 0.8985 | 0.8818 | 1.2655 |
| Playground v2.5 (Li et al., 2024b) | 0.7932 | 0.4442 | 1.3425 | 0.9201 | 0.9146 | 0.8978 | **1.3255** |
| Attention Refocus (Phung et al., 2024) | 0.7058 | 0.4917 | 2.6908 | 0.8208 | 0.9365 | 0.8454 | 0.1869 |
| SLD (Wu et al., 2024) | 0.7115 | 0.4067 | 1.8142 | 0.8463 | 0.8582 | 0.8230 | 0.5190 |
| Layoutgpt (Feng et al., 2023) | 0.7097 | 0.5208 | 2.9092 | 0.8192 | 0.9504 | 0.8505 | 0.0187 |
| MaskUnet (Wang et al., 2025) | 0.6709 | 0.3033 | 2.2950 | 0.8688 | 0.8520 | 0.8180 | 1.1658 |
| Counting Guidance (Kang et al., 2025) | 0.6649 | 0.2575 | 2.1950 | 0.8701 | 0.8527 | 0.8224 | 0.6503 |
| IoCo (Zafar et al., 2026) | 0.6444 | 0.2967 | 2.3467 | 0.8011 | 0.9616 | 0.8505 | 0.0413 |
| CountGen (Binyamin et al., 2025) | 0.7678 | 0.4633 | 1.5883 | 0.8773 | 0.9555 | 0.8952 | 0.9542 |
| SDXL (Podell et al., 2024) | 0.6413 | 0.2958 | 3.1375 | 0.7928 | 0.9588 | 0.8390 | 0.9445 |
| SD v3 (Esser et al., 2024) | 0.8353 | 0.5058 | 0.9492 | 0.9149 | 0.9606 | 0.9260 | 1.2573 |
| Ours (SDXL) | 0.8528 | 0.6125 | 0.9575 | 0.9215 | **0.9714** | 0.9345 | 1.0414 |
| Ours (SD v3) | **0.8920** | **0.6225** | **0.6392** | **0.9528** | 0.9604 | **0.9491** | 1.2681 |

deviations. The aggregation of $\mathcal{L}_{drl}$ and $\mathcal{L}_{ssl}$ follows an identical weighting mechanism.

The overall objective is defined as:

$$\mathcal{L}_{total} = \lambda_1 \mathcal{L}_{acl} + \lambda_2 \mathcal{L}_{drl} + \lambda_3 \mathcal{L}_{ssl}. \qquad (14)$$

where $\lambda_n$ are scalar hyperparameters that control the relative contributions of the individual loss terms. In the *single-category object counting* setting, we set $\lambda_3 = 0$.

**Training-free Guidance.** At each denoising timestep $t$, the latent variable $z_t$ is updated by descending the gradient of the total loss:

$$z_t^+ \leftarrow z_t - \eta_t \nabla_{z_t} \mathcal{L}_{total}. \qquad (15)$$

where $\eta_t$ is the optimization step size.

## 4. Experiments

### 4.1. Experimental Setup

**Implementation Details.** The proposed method can be seamlessly integrated into any diffusion-based text-to-image model. In our implementation, we adopt SDXL (Podell et al., 2024) and Stable Diffusion 3.0 (Esser et al., 2024) as the base models. ***Detailed hyperparameter settings are provided in the appendix.***

**Baselines.** We compare our method with five strong **diffusion-based T2I models**, namely **FLUX.1** (Labs, 2024), **PixArt–Σ** (Chen et al., 2024), **Playground v2.5** (Li et al., 2024b), **Stable Diffusion XL v1.0 (SDXL)** (Podell et al., 2024), and **Stable Diffusion 3.0 (SD v3)** (Esser et al., 2024), as well as four state-of-the-art **object count control methods**, namely **Attention Refocus** (Phung et al., 2024), **SLD** (Wu et al., 2024), **Layoutgpt** (Feng et al., 2023), **MaskUnet** (Wang et al., 2025), **Counting Guidance** (Kang et al., 2025), **IoCo** (Zafar et al., 2026), and **CountGen** (Binyamin et al., 2025).

*Table 2.* Quantitative results on the **T2I-Count** benchmark.

| Method | STA↑ | Acc↑ | MAE↓ | IR↑ |
|---|---|---|---|---|
| FLUX.1 | 0.7567 | 0.4089 | 1.2178 | 0.8828 |
| PixArt-Σ | 0.6945 | 0.3178 | 1.5956 | 0.9672 |
| Playground v2.5 | 0.7191 | 0.3778 | 1.5444 | **1.0089** |
| Attention Refocus | 0.6712 | 0.4489 | 2.5422 | 0.0628 |
| SLD | 0.6398 | 0.3333 | 1.9733 | 0.0805 |
| Layoutgpt | 0.6617 | 0.4756 | 3.1022 | -0.2275 |
| MaskUnet | 0.6148 | 0.2733 | 2.3089 | 0.9549 |
| Counting Guidance | 0.6094 | 0.2467 | 2.1778 | 0.1905 |
| IoCo | 0.7739 | 0.3622 | 1.3689 | 0.4509 |
| CountGen | 0.6879 | 0.4000 | 1.8533 | 0.5059 |
| SDXL | 0.6226 | 0.2978 | 2.2756 | 0.6708 |
| SD v3 | 0.7821 | 0.4444 | 1.1178 | 0.8845 |
| Ours (SDXL) | 0.8123 | 0.5556 | 0.9889 | 0.7248 |
| Ours (SD v3) | **0.8577** | **0.6022** | **0.7644** | 0.9355 |

**Benchmarks.** We evaluate our method across both single-category and multi-category object counting tasks. For *single-category object counting*, we adopt two established benchmarks. (1) **COCOCount** (Binyamin et al., 2025) consists of 200 prompts with object counts ranging from 2 to 10. Each prompt follows the template "A photo of $N$ objects", with object categories sampled from MSCOCO (Lin et al., 2014). (2) **T2I-Count** (Huang et al., 2025), a subset of T2I-CompBench++, covers object counts from 1 to 8. Its object categories extend beyond those in MSCOCO, and the prompts are automatically generated using large language models. For *multi-category object counting*, we construct a new benchmark, **MultiCount**. Prompts are generated using GPT-5 to avoid implausible object pairings. Each prompt contains two object categories with a total object count ranging from 3 to 10. For each count value, we sample 15 prompts, resulting in a total of 120 prompts.

**Evaluation Metrics.** We evaluate object counting performance using multiple complementary metrics. **Accu-**

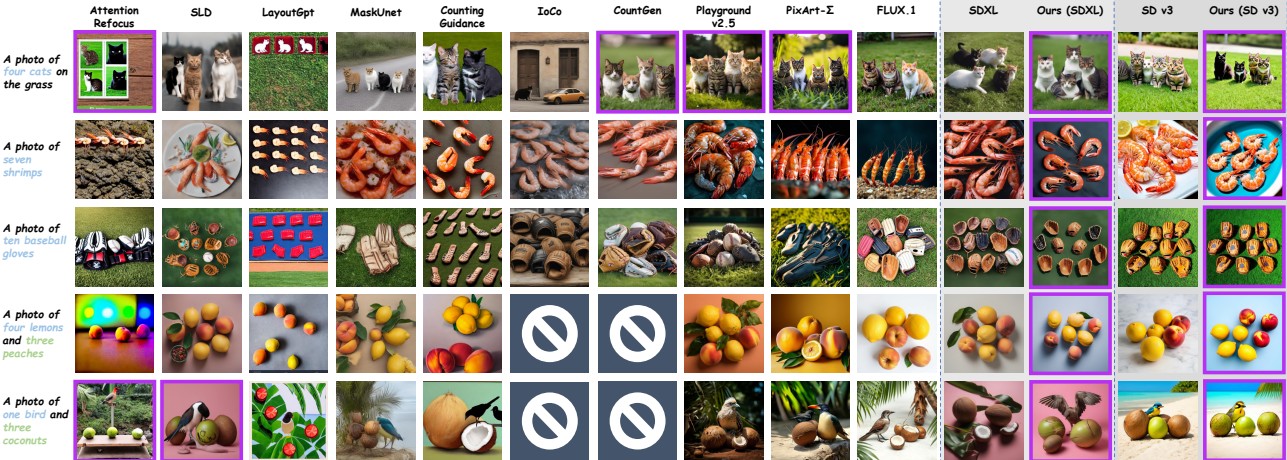

*Figure 3.* Qualitative comparison of baseline methods and our approach. Images outlined in purple contain the correct number of objects.

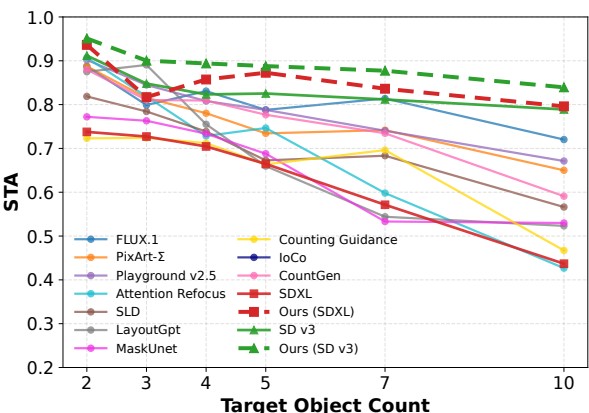

*Figure 4.* Comparison of STA performance between our method and baseline approaches as the target object count increases.

*Table 3.* Quantitative results on the **MultiCount** benchmark.

| Method | STA↑ | Acc↑ | MAE↓ | IR↑ |
|---|---|---|---|---|
| FLUX.1 | 0.7441 | 0.2292 | 0.9347 | 1.4826 |
| PixArt-Σ | 0.5563 | 0.0722 | 1.6132 | 1.3353 |
| Playground v2.5 | 0.6245 | 0.1222 | 1.4028 | 1.4480 |
| Attention Refocus | 0.4685 | 0.1028 | 1.9486 | -0.0678 |
| SLD | 0.4980 | 0.0986 | 1.8799 | 0.9665 |
| Layoutgpt | 0.4619 | 0.1125 | 2.0500 | 0.0459 |
| MaskUnet | 0.3885 | 0.0250 | 2.2062 | 1.3425 |
| Counting Guidance | 0.3694 | 0.0194 | 2.2299 | 0.3603 |
| SDXL | 0.5550 | 0.0833 | 1.8882 | 1.2096 |
| SD v3 | 0.6569 | 0.1486 | 1.2236 | 1.4238 |
| Ours (SDXL) | 0.7060 | 0.2542 | 1.0965 | 1.3620 |
| Ours (SD v3) | **0.7903** | **0.3500** | **0.7514** | **1.4945** |

racy (Acc) measures the proportion of prompts for which the generated object count exactly matches the target object count, while **MAE** (Mean Absolute Error) quantifies the average magnitude of the discrepancy between the target and generated counts. **Precision** and **Recall** penalize over-generation and under-generation, respectively, and are jointly summarized by the **F1-score**. To assess perceptual quality, we additionally report **ImageReward (IR)** (Xu et al., 2023). We further introduce **Saturated Truncated Accuracy (STA)** for robust counting evaluation: $\text{STA} = \frac{1}{|\mathcal{D}|} \sum_{i=1}^{|\mathcal{D}|} \left(1 - \min\left(\frac{|\hat{y}_i - y_i|}{y_i}, 1\right)\right)$, where $y_i$ and $\hat{y}_i$ denote the ground-truth count specified in the $i$-th textual prompt and the number of objects detected in the generated image, respectively, and $|\mathcal{D}|$ denotes the total number of evaluated samples. STA measures relative counting error while truncating extreme deviations. Object instances are detected using **Grounding DINO** (Liu et al., 2024).

## 4.2. Quantitative Results

**Comparison with Baselines.** Table 1 and Table 2 report single-category object counting results on the COCOCount and T2I-Count benchmarks, respectively. Several observations can be made: (1) Our method substantially improves all evaluation metrics over the base models (SDXL and SD v3), while maintaining comparable IR scores, showing that enhanced counting does not compromise image quality. (2) Despite the relatively poor performance of the vanilla SDXL model, our SDXL-based method achieves second-place performance, demonstrating strong generalization across different base models. (3) Built on SD v3, our method outperforms all baselines on all metrics except IR, where Playground v2.5 achieves the highest score due to advanced image quality techniques. Table 3 presents multi-category object counting results on the MultiCount benchmark. Trends are similar to the single-category setting, with one exception: FLUX.1 outperforms SD v3 and SDXL. However, augmenting SD v3 with our method surpasses FLUX.1, highlighting its effectiveness in more challenging

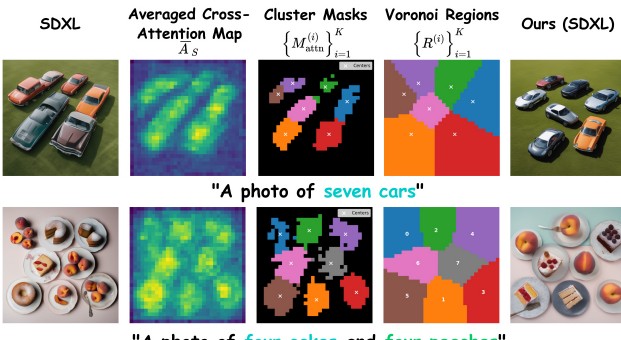

*Figure 5.* Visualization of the object region planning. From left to right: images generated by the base model SDXL, the timestep-averaged cross-attention map of the specific object token, the cluster masks, the Voronoi regions and the final images generated by our method.

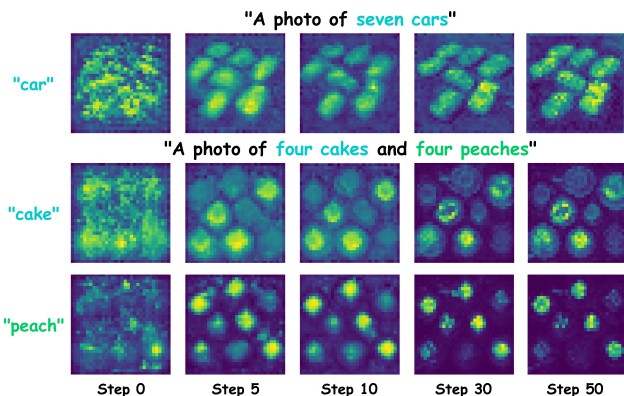

*Figure 6.* Visualization of cross-attention evolution under the attention-guided layout consistency strategy.

*Table 4.* Quantitative ablation results on the **T2I-Count** benchmark.

| Ablation | STA↑ | Acc↑ | MAE↓ | IR↑ |
|---|---|---|---|---|
| w/o OSE | 0.7886 | 0.5378 | 1.1267 | 0.7085 |
| w/o LRR + OSE | 0.7721 | 0.5022 | 1.2733 | 0.6819 |
| w/o $\mathcal{L}_{acl}$ | 0.6701 | 0.3622 | 2.0422 | 0.6904 |
| w/o $\mathcal{L}_{drl}$ | 0.8043 | 0.5311 | 1.1644 | 0.7074 |
| w/o $\mathcal{L}_{acl} + \mathcal{L}_{drl}$ | 0.6946 | 0.3511 | 1.7733 | 0.7168 |
| Ours (SDXL) | **0.8123** | **0.5556** | **0.9889** | **0.7248** |

multi-category object counting tasks. Overall, these results validate the proposed method across diverse counting scenarios. All methods use the same random seeds for fair comparison.

**Effect of Object Count on Performance.** To study how counting accuracy varies with the target object count, we compare our method with baselines on the COCOCount benchmark. We generate six images per prompt, resulting in a total of 1,200 samples. As illustrated in Figure 4, our SD v3-based method consistently outperforms all baselines across different object counts, while the SDXL-based variant demonstrates competitive performance.

### 4.3. Qualitative Analysis

**Qualitative Evaluation.** Figure 3 presents a qualitative comparison across both single-category and multi-category object counting tasks. In the single-category setting, while diffusion-based T2I models achieve reasonable success with small target counts, performance degrades significantly as the count increases. Although existing count-control baselines offer improvements in low-count scenarios, their reliability remains inconsistent across diverse prompts. In the more challenging multi-category setting, most baselines not only produce incorrect object counts but also fail to generate all object categories specified in the prompt. In contrast,

our method consistently generates the precise number of instances for each category while ensuring all specified objects are present. These results underscore the robust controllability of our approach in complex counting and multi-category composition scenarios.

**Visualization of Object Region Planning.** We visualize the intermediate outputs of the proposed *Object Region Planning*. As shown in Figure 5, the top and bottom rows correspond to single-category and multi-category object counting, respectively. The visualizations show that our method generates region maps that accurately designate object locations according to the target object count. These regions serve as structural priors during denoising, ensuring well-separated, semantically coherent object instances. Moreover, the region maps closely align with the final synthesized images: objects appear in the correct count and within their designated regions.

**Evolution of Cross-Attention Maps.** We analyze the temporal evolution of cross-attention maps under the proposed *Attention-guided Layout Consistency Strategy*. As shown in Figure 6, our method effectively regulates denoising, ensuring each planned region produces a single coherent object. This consistency holds for both single-category and multi-category object counting, mitigating common failure modes such as object merging.

### 4.4. Ablation Studies

We ablate each module on the T2I-Count benchmark to evaluate its impact. The variants are: (1) w/o OSE: without *Object Saliency Enhancement*; (2) w/o LRR + OSE: without the *Count-Aware Noise Adjustment Strategy*; (3) w/o $\mathcal{L}_{acl}$: without *Attention Concentration Loss*; (4) w/o $\mathcal{L}_{drl}$: without *Dispersion Regularization Loss*; (5) w/o $\mathcal{L}_{acl} + \mathcal{L}_{drl}$: without the *Attention-Guided Layout Consistency Strategy*. Results are reported in Table 4.

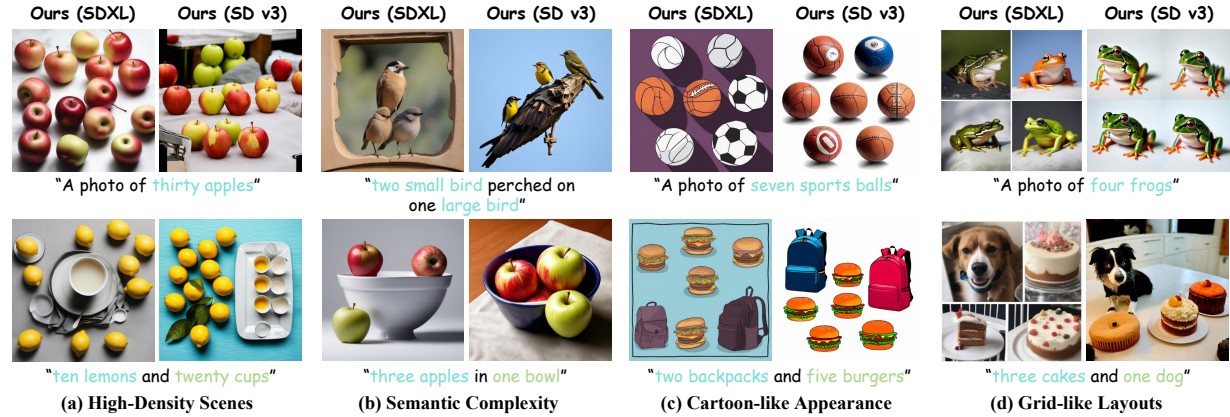

*Figure 7.* Failure cases and limitations of the proposed method: (a) High-Density Scenes, (b) Semantic Complexity, (c) Cartoon-like Appearances and (d) Grid-like Layouts.

**Effect of Count-Aware Noise Adjustment.** Removing *Object Saliency Enhancement* (w/o OSE) leads to noticeable performance degradation across all counting metrics, indicating its role in sharpening region-wise responses and stabilizing object formation. Removing both LRR and OSE (w/o LRR + OSE) further degrades performance, confirming that the count-aware noise adjustment strategy is essential for aligning early layouts with the target object count.

**Effect of Attention-Guided Layout Consistency.** Removing $\mathcal{L}_{acl}$ causes the largest drop, highlighting the importance of compact attention to prevent ambiguous layouts. Removing $\mathcal{L}_{drl}$ also degrades performance, though less severely, showing that suppressing dispersed attention complements attention concentration. Removing both losses (w/o $\mathcal{L}_{acl}$ + $\mathcal{L}_{drl}$) causes substantial performance drops, indicating that these two losses work synergistically to enforce consistent and well-separated attention patterns during denoising.

Overall, these results confirm that each component is vital to simultaneously enhancing object counting accuracy and the visual fidelity of the generated images.

### 4.5. Limitations and Future Work

Despite significantly enhancing the object counting capabilities of text-to-image diffusion models, as shown in Figure 7, our method still has several limitations, which point to promising directions for future research.

**Scalability to High-Density Scenes.** While our method performs reliably for moderate object counts, it faces increasing difficulty when scaling to scenes with a large number of objects (e.g., $N > 30$). This limitation mainly stems from the finite spatial resolution of the initial noise latent and the fixed representational capacity of the backbone diffusion models. As object density increases, the planned Voronoi regions become progressively compressed, leading to semantic crowding, where adjacent region boundaries are no longer well distinguished. As a result, objects may merge or be under-generated (see Figure 7(a)).

**Sensitivity to Prompt Semantic Complexity.** Although our method is robust to linguistic variations, it may still struggle with prompts that involve complex inter-object relationships (e.g., "two small birds perched on one large bird"). In such cases, the non-overlapping nature of Voronoi-based layout planning may struggle to accommodate complex semantic hierarchies where objects are meant to overlap or be nested (see Figure 7(b)).

**Trade-off Between Control and Visual Realism.** The proposed *Attention-Guided Layout Consistency Strategy* may occasionally produce stylized, cartoon-like appearances or rigid, grid-like layouts (see Figure 7(c) and (d)). This behavior arises because rigid attention constraints may override the model's intrinsic generative priors. By strictly enforcing attention within predefined regions, the model may prioritize spatial compliance at the expense of fine-grained texture, lighting, and shape coherence required for high-fidelity photorealism.

## 5. Conclusion

This work identifies the Noise-Induced Layout Prior as a primary bottleneck for precise object counting in text-to-image diffusion models. Our analysis demonstrates that initial noise configurations dominate early-stage denoising, often overriding prompt-specified counts. To mitigate this, we proposed a novel training-free framework that intervenes in object arrangement during the early stages of denoising. Extensive experiments demonstrate that our approach achieves accurate and controllable object counting while preserving high image fidelity. These findings highlight the critical role of early-stage noise and attention dynamics, offering a promising path for controlling complex compositional properties in generative models.

## Acknowledgements

This work was supported in part by the National Natural Science Foundation of China (Grant No. 62471168 and 62422204), in part by the Zhejiang Provincial Natural Science Foundation of China (Grant No. LRG26F020001 and LMS26F020015), in part by Major Research Program of the Zhejiang Provincial Natural Science Foundation (Grant No. LD24F020015), in part by the Key Research and Development Program of Zhejiang Province (Grant No. 2025C01026) and in part by the Scientific Research Innovation Capability Support Project for Young Faculty.

## Impact Statement

This paper presents work whose goal is to advance the field of Machine Learning. There are many potential societal consequences of our work, none which we feel must be specifically highlighted here.

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

# A. Further Details of the Proposed Method

## A.1. Count-Aware Noise Adjustment Strategy

**Object Region Planning.** Given a prompt like "A photo of three apples on the ground", we first perform $S$ denoising steps using a T2I diffusion model to obtain a timestep-averaged cross-attention map $\bar{A}_S \in \mathbb{R}^{H \times W}$ corresponding to a specific object token (e.g., "apple"). We use the min-max normalization to make the attention map normalized to the range $[0, 1]$ as follows:

$$\hat{A}_S = \frac{\bar{A}_S - \min(\bar{A}_S)}{\max(\bar{A}_S) - \min(\bar{A}_S)}. \tag{16}$$

Next, we generate a binary high-response attention mask $M_{\text{attn}}$ and apply the K-means clustering algorithm to the coordinates within $M_{\text{attn}}$, using $\hat{A}_S$ as attention-weighted distances to guide the clustering. This yields $K$ cluster centers $\{c^{(i)}\}_{i=1}^K$ and corresponding cluster masks $\{M_{\text{attn}}^{(i)}\}_{i=1}^K$, where $K$ is the target object count:

$$\{c^{(i)}\}_{i=1}^K, \{M_{\text{attn}}^{(i)}\}_{i=1}^K = \text{Cluster}(M_{\text{attn}}, \hat{A}_S, K). \tag{17}$$

To ensure that each object instance occupies a distinct region, we construct a Voronoi diagram over the latent space based on the previously obtained cluster centers, partitioning it into $K$ mutually exclusive regions $\{R^{(i)}\}_{i=1}^K$. Let $\Omega \subset \mathbb{Z}^2$ denote the discrete $H \times W$ latent space. Each object region is defined as:

$$R^{(i)} = \left\{ p \in \Omega \,\middle|\, \arg\min_j \|p - c^{(j)}\| = i \right\}. \tag{18}$$

To handle prompts with multiple object categories (e.g., "two apples and three peaches"), we merge the object-specific attention maps via element-wise maximization:

$$\hat{A}_S(x, y) = \max_{n \in \mathcal{N}} \hat{A}_{S,n}(x, y), \quad K = \sum_{n \in \mathcal{N}} K_n. \tag{19}$$

where $\mathcal{N}$ denotes the set of object categories specified in the prompt, $\hat{A}_{S,n}$ is the normalized cross-attention map corresponding to object category $n$, $(x, y)$ is the spatial position in the cross-attention map, and $K_n$ denotes its target object count. $\hat{A}_S$ captures the union of spatial regions favored for object generation, while $K$ represents the total number of object instances to be generated. Additionally, we assign each region $R^{(i)}$ to object category $n$ according to its target object count $K_n$ by comparing the attention responses $\hat{A}_{S,n}$ within the region. Specifically, for each region, we quantify the response strength of each object by aggregating its normalized attention values over $R^{(i)}$, and assign the region to the object that exhibits the highest overall activation.

**Latent Representation Reorganization.** Based on the previously computed cross-attention map $\bar{A}_S \in \mathbb{R}^{H \times W}$, we perform region-wise min–max normalization to obtain a set of normalized attention maps $\hat{A}_S^{(i)}$ corresponding to each object region $R^{(i)}$:

$$\hat{A}_S^{(i)} = \frac{\bar{A}_S^{(i)} - \min(\bar{A}_S^{(i)})}{\max(\bar{A}_S^{(i)}) - \min(\bar{A}_S^{(i)})}. \tag{20}$$

Here, $\bar{A}_S^{(i)}$ represents the cross-attention map of region $R^{(i)}$. $\min(\bar{A}_S^{(i)})$ and $\max(\bar{A}_S^{(i)})$ are the minimum and maximum attention values within region $R^{(i)}$, respectively.

Then, we reorganize the latent feature vectors within each region $R^{(i)}$ by adjusting their spatial positions based on their attention-based significance, encouraging highly attended vectors to move closer to the region center $c^{(i)}$. Let $P^{(i)} = \{p_1^{(i)}, p_2^{(i)}, \ldots, p_N^{(i)}\}$ denote the set of all spatial positions within region $R^{(i)}$, where $N$ is the total number of spatial positions contained in $R^{(i)}$ and $p_j^{(i)} = (h_j, w_j)$ represents a discrete coordinate. These positions are then sorted in ascending order based on their Euclidean distance to the region center $c^{(i)}$:

$$\mathcal{D}^{(i)} = \text{argsort}_{p \in P^{(i)}} \left( \|p - c^{(i)}\|_2 \right) \quad (\uparrow). \tag{21}$$

producing an ordered list $\{p_{[1]}^{(i)}, p_{[2]}^{(i)}, \ldots, p_{[N]}^{(i)}\}$, where $p_{[1]}^{(i)}$ is the spatial position closest to the region center $c^{(i)}$. Next, we sort the same set of positions by their attention values in descending order:

$$\mathcal{A}^{(i)} = \text{argsort}_{p \in P^{(i)}} \left( \hat{A}_S^{(i)}(p) \right) \quad (\downarrow). \tag{22}$$

producing another ordered list $\{q_{[1]}^{(i)}, q_{[2]}^{(i)}, \ldots, q_{[N]}^{(i)}\}$, where $q_{[1]}^{(i)}$ is the position with the highest attention score.

Finally, we reorganize the latent representation by relocating each latent vector from its original high-attention position to a spatial position closer to the region center. Specifically, the latent vector at $q_{[j]}^{(i)}$ is reassigned to $p_{[j]}^{(i)}$:

$$z_T^*(p_{[j]}^{(i)}) \leftarrow z_T(q_{[j]}^{(i)}), \quad \text{for } j = 1, 2, \ldots, N. \tag{23}$$

where $z_T(p)$ denotes the latent vector at spatial coordinate $p$. By applying the above reorganization across all object regions, we obtain the updated noise representation $z_T^*$.

## A.2. Attention-Guided Layout Consistency Strategy

**Centroid Computation.** For each region $R^{(i)}$, we compute the attention-weighted centroid $c_t^{(i)} = (x_t^{(i)}, y_t^{(i)})$ by taking the weighted average of spatial coordinates, where the

weights are given by $\tilde{A}_t^{(i)}$. Formally, for $(x, y) \in R^{(i)}$:

$$
\begin{cases}
x_t^{(i)} = \dfrac{\sum_{(x,y) \in R^{(i)}} \tilde{A}_t^{(i)}(x, y) \cdot x}{\sum_{(x,y) \in R^{(i)}} \tilde{A}_t^{(i)}(x, y)}, \\[4mm]
y_t^{(i)} = \dfrac{\sum_{(x,y) \in R^{(i)}} \tilde{A}_t^{(i)}(x, y) \cdot y}{\sum_{(x,y) \in R^{(i)}} \tilde{A}_t^{(i)}(x, y)}.
\end{cases}
\tag{24}
$$

**Category-Aware Latent Control.** To handle prompts containing multiple object categories, we extend our latent optimization mechanism to enable category-specific layout control. Let $\mathcal{N}$ denote the set of object categories specified in the prompt, where each category $n \in \mathcal{N}$ is associated with a target object count $K_n$.

Starting from the noise-adjusted latent $z_T'$ obtained via the count-aware noise adjustment strategy, we duplicate it into a set of category-specific latent branches $\{z_{T,n}\}_{n \in \mathcal{N}}$. These duplicated latents share the same initial noise but are used to separately analyze and regulate object-specific attention responses during denoising, facilitating category-aware region assignment and optimization.

Meanwhile, the latent space is partitioned into a set of candidate spatial regions $\{R^{(i)}\}_{i=1}^K$ via object region planning, where $K = \sum_{n \in \mathcal{N}} K_n$. These regions are object-agnostic at initialization and are dynamically assigned to specific object categories during the denoising process. Specifically, at each denoising timestep $t$, we extract the cross-attention map $A_{t,n}$ for each object category $n$ and apply min–max normalization (Eq. 16), yielding the collection $\{\hat{A}_{t,n}\}_{n \in \mathcal{N}}$ over all object categories.

For a given region $R^{(i)}$, the attention response of object category $n$ is computed as

$$
\text{Score}_{t,n}^{(i)} = \sum_{(x,y) \in R^{(i)}} \hat{A}_{t,n}(x, y).
\tag{25}
$$

which measures the overall attention mass of object category $n$ within region $R^{(i)}$.

Region–category correspondences are formulated as a global assignment problem with explicit count constraints. At each denoising timestep, we construct a bipartite graph between object categories and candidate regions, where edge weights are defined by the attention response scores $\{\text{Score}_{t,n}^{(i)}\}$. Each category $n$ is allowed to match with at most $K_n$ regions, and each region can be assigned to at most one category. The optimal assignment is obtained using the *Hungarian algorithm* (Kuhn, 2010), which maximizes the total attention response over all selected region–category pairs. The assignment is recomputed dynamically at every denoising timestep, ensuring count-aware and object-consistent region–category alignment throughout the entire denoising process.

Based on the established correspondence, each category-specific latent branch is optimized independently. Specifically, for latent $z_{t,n}$, we apply the *Attention-Guided Layout Consistency Strategy* only within its assigned regions $\mathcal{R}_n$, regulating the evolution of the corresponding cross-attention map $\hat{A}_{t,n}$ while leaving other regions unaffected.

Finally, the optimized latent branches are merged to form the latent for the next denoising step. The merged latent $z_t$ is constructed by spatially combining all branches according to their assigned regions:

$$
z_t = \sum_{n \in \mathcal{N}} \sum_{R^{(i)} \in \mathcal{R}_n} \mathbf{1}_{R^{(i)}} \odot z_{t,n}.
\tag{26}
$$

where $\mathbf{1}_{R^{(i)}}$ denotes the binary mask of region $R^{(i)}$.

On the other hand, in multi-category object counting, Otsu's method becomes unreliable. For instance, the cross-attention map of a specific object token (e.g., "apple") often contains unintended responses in regions belonging to other object categories (e.g., "peach"). Because Otsu's method relies on a global threshold, these interfering signals obscure the separation of the target object from the background. To address this issue, we compute $\tilde{A}_t^{(i)}(x, y)$ using an adaptive threshold $\tau_k$, defined as the $k$-th percentile of the normalized attention map $\hat{A}_t^{(i)}$, such that only the Top-$k\%$ highest attention values (with $k = 45$ in our implementation) are retained for subsequent computation. Formally, for $(x, y) \in R^{(i)}$:

$$
\tilde{A}_t^{(i)}(x, y) = \begin{cases} \hat{A}_t^{(i)}(x, y), & \text{if } \hat{A}_t^{(i)}(x, y) \geq \tau_k, \\ 0, & \text{otherwise.} \end{cases}
\tag{27}
$$

## B. Additional Experimental Setup

### B.1. Implementation Details

We evaluate our method under two distinct architectural settings. For UNet-based architectures, we build upon Stable Diffusion XL (SDXL) v1.0 and conduct all experiments on a single NVIDIA RTX 4090 GPU. In the object region planning module, we set the number of denoising steps to $S = 20$ and the attention threshold to $\tau = 0.6$. For object saliency enhancement, the modulation weights are set to $\gamma = 1.1$ and $\beta = 0.975$. In the attention-guided layout consistency strategy, optimization is performed over the first four denoising steps, with a scaling factor of $\lambda_o = 1.8$ and a temperature coefficient of $\lambda_s = 5$. Additionally, we set $\lambda_d = 0.05$ in $\mathcal{L}_{\text{acl}}$. For the **COCOCount** benchmark, the loss weights are set to $\lambda_1 = 1$, $\lambda_2 = 15$, and $\lambda_3 = 0$. For the **T2I-Count** benchmark, we use $\lambda_1 = 1$, $\lambda_2 = 12$, and $\lambda_3 = 0$. For the **MultiCount** benchmark, the loss weights are set to $\lambda_1 = 1$, $\lambda_2 = 15$, and $\lambda_3 = 80$. In addition, extra optimization is applied at the 10th and 20th denoising steps to further enhance the effect of the *Semantic Suppression Loss*, using $\lambda_1 = 0$, $\lambda_2 = 0$, and $\lambda_3 = 80$.

For transformer-based architectures, we build upon Stable Diffusion 3 (SD v3) and conduct experiments on a single NVIDIA A800 GPU. The attention threshold is adjusted to $\tau = 0.5$. For the **COCOCount** and **T2I-Count** benchmarks, the loss weights are set to $\lambda_1 = 0.2$, $\lambda_2 = 10$, and $\lambda_3 = 0$. For the **MultiCount** benchmark, the loss weights are set to $\lambda_1 = 0.2$, $\lambda_2 = 10$, and $\lambda_3 = 120$. In addition, extra optimization is applied at the 10th denoising steps using $\lambda_1 = 0$, $\lambda_2 = 0$, and $\lambda_3 = 120$. All other hyperparameters remain identical to those in the UNet-based configuration. For both architectures, all experiments are conducted using a fixed set of randomly sampled seeds to ensure fairness and reproducibility.

### B.2. Benchmarks

**COCOCount Benchmark.** The COCOCount benchmark (Binyamin et al., 2025) comprises six target object count categories: 2, 3, 4, 5, 7, and 10. The categories with counts of 2 and 3 each contain 34 prompts, while the remaining categories contain 33 prompts each, resulting in a total of 200 prompts.

**MultiCount Benchmark.** We construct a new benchmark, MultiCount, to evaluate multi-category object counting, where each prompt contains two distinct object categories with specified counts. MultiCount includes 26 object categories: *bird, cat, dog, frog, horse, sheep, bear, apple, lemon, orange, peach, coconut, donut, cake, burger, bottle, egg, cup, vase, clock, lamp, phone, camera, backpack, car, kite*. These categories are chosen to align with commonly used object classes in existing text-to-image counting benchmarks, such as T2I-Count (Huang et al., 2025) and COCOCount (Binyamin et al., 2025). They span animals, food items, and everyday objects that are visually distinctive and frequently appear in natural image compositions. We use prompts of the form "a photo of $N_1$ object1 and $N_2$ object2", where $N_1, N_2 \in \{1, \ldots, 9\}$ and $K = N_1 + N_2 \in \{3, \ldots, 10\}$. For each value of $K$, we construct 15 prompts, resulting in a total of 120 prompts. Implausible object pairs are filtered using GPT-5 in our MultiCount benchmark.

### B.3. Baselines

**FLUX.1** (Labs, 2024). We used the *black-forest-labs/FLUX.1-schnell* checkpoint from Hugging Face. Images are generated using 4 sampling steps at a resolution of $1024 \times 1024$.

**PixArt-$\Sigma$** (Chen et al., 2024). We used the *PixArt-alpha/PixArt-Sigma-XL-2-1024-MS* checkpoint from Hugging Face and followed the official guidance using the `diffusers` library. Images are generated using 20 sampling steps at a resolution of $1024 \times 1024$.

**Playground v2.5** (Li et al., 2024b). We used the *playgroundai/playground-v2.5-1024px-aesthetic* checkpoint from Hugging Face and followed the official guidance using the `diffusers` library. Images are generated using 50 sampling steps at a resolution of $1024 \times 1024$.

**Stable Diffusion XL v1.0 (SDXL)** (Podell et al., 2024). We used the *stabilityai/stable-diffusion-xl-base-1.0* checkpoint from Hugging Face. Images are generated using 50 sampling steps at a resolution of $1024 \times 1024$.

**Stable Diffusion 3.0 (SD v3)** (Esser et al., 2024). We used the official *stabilityai/stable-diffusion-3-medium* checkpoint from Hugging Face. Images are generated with 50 sampling steps at a resolution of $1024 \times 1024$.

**Attention Refocus** (Phung et al., 2024). We used the official implementation available at `https://github.com/Attention-Refocusing/attention-refocusing`. Following the provided prompt templates, we used GPT-4 to generate bounding boxes for the prompts in our evaluation datasets.

**SLD** (Wu et al., 2024). We used the official implementation available at `https://github.com/tsunghan-wu/SLD`. All settings followed the default configuration provided in the repository.

**Layoutgpt** (Feng et al., 2023). We used the official implementation available at `https://github.com/weixi-feng/LayoutGPT`. All settings followed the default configuration provided in the repository.

**MaskUnet** (Wang et al., 2025). We used the official implementation available at `https://github.com/gudaochangsheng/MaskUnet`. All settings followed the default configuration provided in the repository.

**Counting Guidance** (Kang et al., 2025). We used the official implementation available at `https://github.com/furiosa-ai/counting-guidance`. All settings followed the default configuration provided in the repository.

**CountGen** (Binyamin et al., 2025). We used the official implementation available at `https://github.com/Litalby1/make-it-count`. All settings followed the default configuration provided in the repository.

**IoCo** (Zafar et al., 2026). We used the official implementation from `https://github.com/ozzafar/count_token_optimization`. All settings followed the default configuration provided in the repository.

### B.4. Evaluation Metrics.

We evaluate object counting performance using a set of complementary metrics. For each sample, let $y_i$ represent the ground-truth count specified in the $i$-th textual prompt,

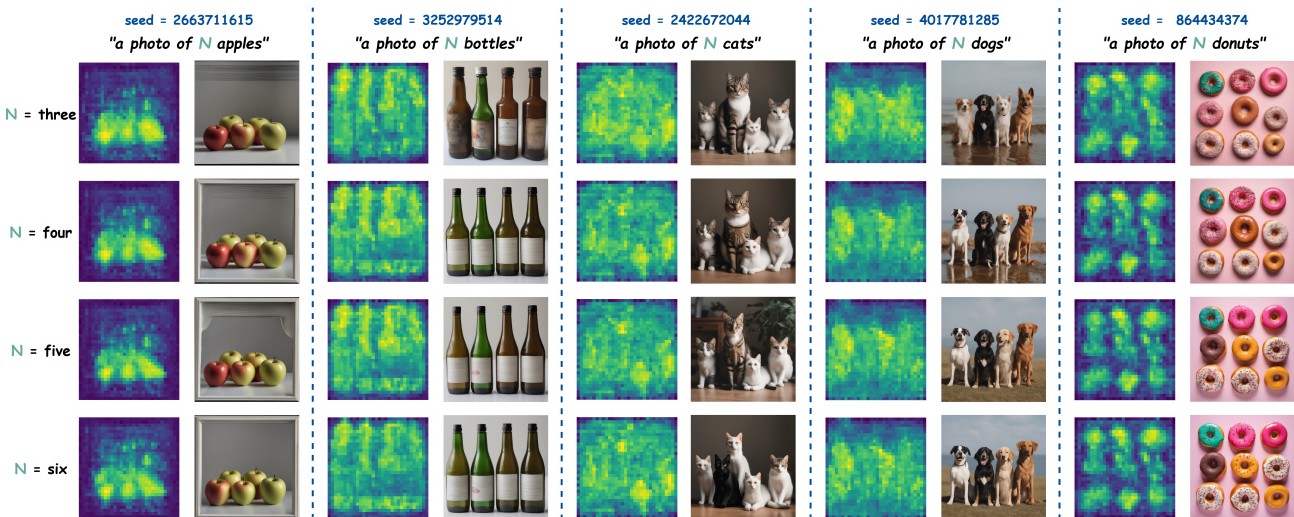

*Figure 8.* Effect of initial noise on spatial layout formation. Additional visualization results for different object categories under a fixed initial noise. Prompts vary only in the target object count. Early-stage cross-attention maps (step = 5) and the corresponding final outputs show highly similar spatial layouts across different counts, suggesting a strong dependence of layout formation on the initial noise.

and let $\hat{y}_i$ denote the number of objects detected in the generated image. The total number of evaluated samples is denoted by $|\mathcal{D}|$.

**Accuracy (Acc)** assesses the frequency of perfect count alignment across the evaluation set $\mathcal{D}$:

$$\text{Acc} = \frac{1}{|\mathcal{D}|} \sum_{i=1}^{|\mathcal{D}|} [\hat{y}_i = y_i]. \tag{28}$$

where $[\cdot]$ is the Iverson bracket.

**Mean Absolute Error (MAE)** quantifies the average magnitude of the discrepancy between the target and generated counts:

$$\text{MAE} = \frac{1}{|\mathcal{D}|} \sum_{i=1}^{|\mathcal{D}|} |\hat{y}_i - y_i|. \tag{29}$$

To characterize over-counting and under-counting behaviors, we adopt **Precision** and **Recall**. For each sample, we define the sample-wise Precision ($\text{P}_i$) and Recall ($\text{R}_i$) as follows:

$$\text{P}_i = \frac{\min(\hat{y}_i, y_i)}{\hat{y}_i}, \quad \text{R}_i = \frac{\min(\hat{y}_i, y_i)}{y_i}. \tag{30}$$

When the predict count $\hat{y}_i = 0$, the sample-wise Precision $\text{P}_i = 0$. The dataset-level Precision and Recall are obtained by averaging these per-sample values over the entire evaluation set $\mathcal{D}$:

$$\text{Precision} = \frac{1}{|\mathcal{D}|} \sum_{i=1}^{|\mathcal{D}|} \text{P}_i, \quad \text{Recall} = \frac{1}{|\mathcal{D}|} \sum_{i=1}^{|\mathcal{D}|} \text{R}_i. \tag{31}$$

**F1-score (F1)** serves as the harmonic mean of the sample-wise Precision and Recall, providing a balanced measure of counting performance:

$$\text{F1}_i = \frac{2 \cdot \text{P}_i \cdot \text{R}_i}{\text{P}_i + \text{R}_i}. \tag{32}$$

When the denominator is zero, the sample-wise F1-score $\text{F1}_i = 0$. Similar to the other metrics, the final F1-score is reported as the average of the sample-wise values over the entire evaluation set $\mathcal{D}$:

$$\text{F1} = \frac{1}{|\mathcal{D}|} \sum_{i=1}^{|\mathcal{D}|} \text{F1}_i. \tag{33}$$

In addition, we introduce **Saturated Truncated Accuracy (STA)** for robust counting evaluation:

$$\text{STA} = \frac{1}{|\mathcal{D}|} \sum_{i=1}^{|\mathcal{D}|} \left( 1 - \min\left( \frac{|\hat{y}_i - y_i|}{y_i}, 1 \right) \right). \tag{34}$$

STA measures relative counting error while truncating extreme deviations.

## C. Analysis of Noise-Induced Layout Prior

### C.1. Additional Visual Analysis

To further illustrate the impact of initial noise on spatial layouts and object counts in T2I diffusion models like SDXL, we present additional visual examples in Figure 8. Each pair of columns corresponds to a specific object category (e.g., "apples"). Within each group, all images are generated using a fixed initial noise (i.e., the same random seed), while the prompts differ only in the specified object count (e.g., "a photo of {three, four, five, six} apples").

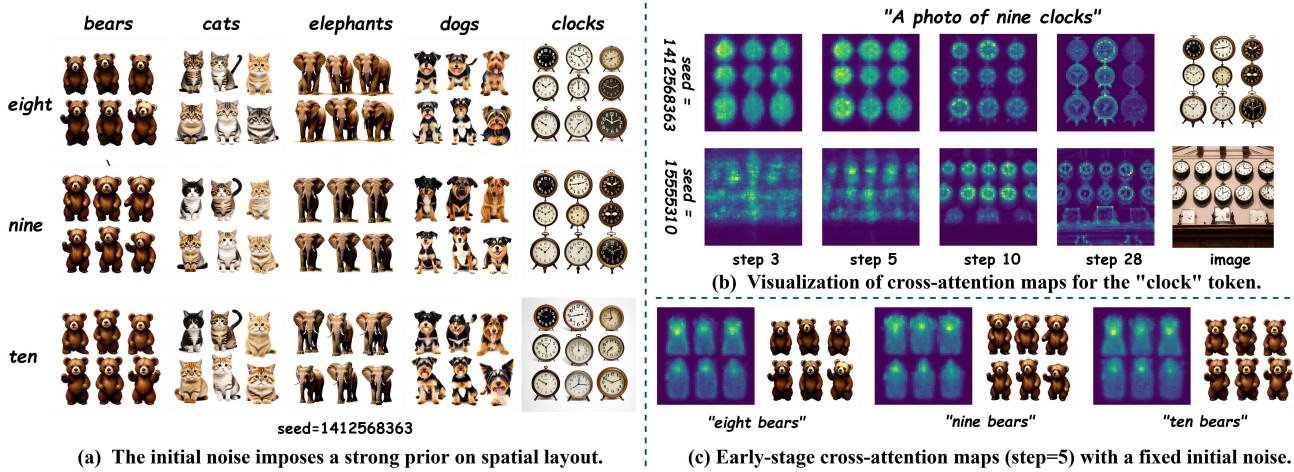

*Figure 9.* Initial noise strongly governs spatial layout and object count in SD v3.

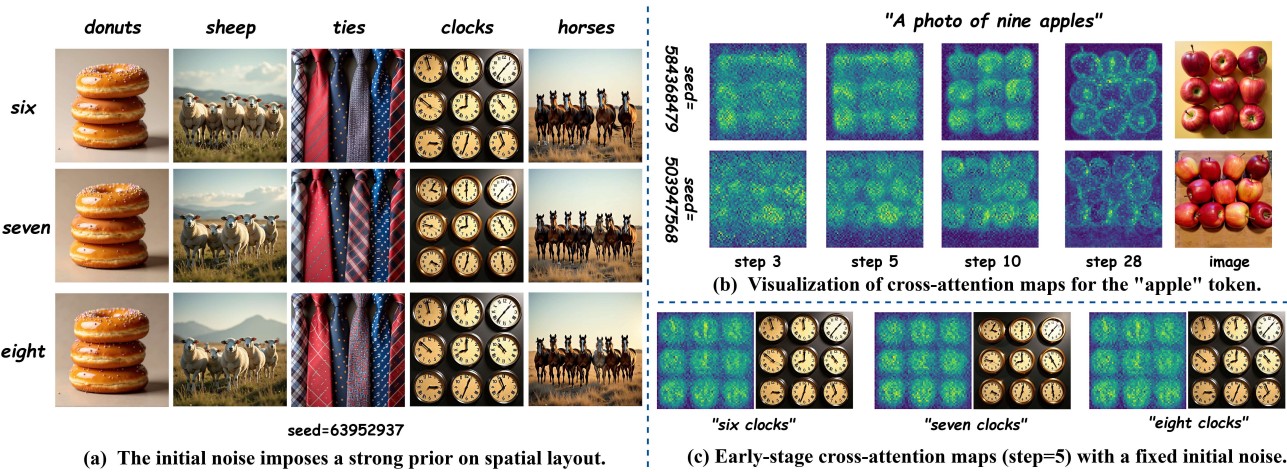

*Figure 10.* Initial noise strongly governs spatial layout and object count in FLUX.1.

For each group, the left column visualizes the early-stage cross-attention map (step = 5) for the corresponding object token, and the right column shows the final generated images. Despite explicit variations in the target object count across prompts, the resulting images exhibit highly consistent spatial arrangements and nearly identical object counts. This invariance indicates that once a coarse object-level layout is established by the initial noise, subsequent text conditioning struggles to alter it.

These observations provide further evidence of a strong *Noise-Induced Layout Prior*, which constrains object-level structure during generation. Such a prior significantly undermines counting accuracy in T2I diffusion models and exposes a fundamental limitation in their ability to satisfy numerical constraints.

More phenomena of *Noise-Induced Layout Prior* can be observed in DiT-based T2I models such as SD v3 and FLUX.1 (see Figures 9 and 10).

## C.2. Quantitative Analysis

As discussed in the main paper, when the initial noise $z_T$ is fixed, varying the target object count $C$ in prompts $\{y_i, y_j\}$ leads to cross-attention maps $\mathbf{A}_t$ that exhibit a persistent and highly similar spatial structure during the early denoising stages, i.e., $t > T - \Delta t$.

To quantitatively measure spatial layout divergence, we employ the Kullback-Leibler (KL) divergence between pairs of cross-attention maps. Specifically, we consider two comparison settings.

**Group 1 (Fixed noise, varying counts).** At each timestep, we compute the KL divergence between cross-attention maps generated from prompts sharing the same object category but specifying different object counts (e.g., "three dogs" vs. "four dogs") under a *fixed* initial noise $z_T$. The reported value is obtained by averaging over all such prompt pairs.

**Group 2 (Different noise, different categories).** At each

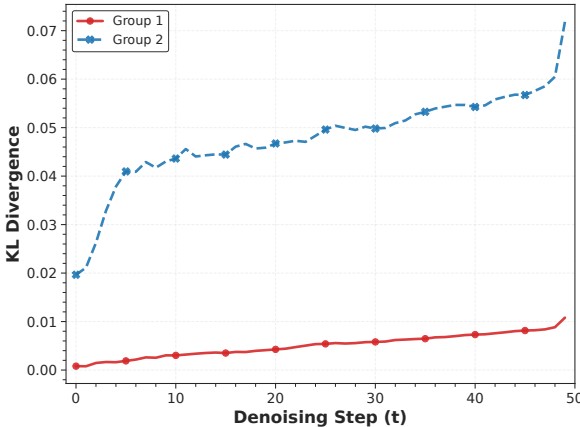

*Figure 11.* Temporal evolution of spatial layout divergence. KL divergence between cross-attention maps for Group 1 (fixed initial noise, varying target object counts) and Group 2 (different initial noise, different object categories) across denoising timesteps. Group 1 exhibits significantly lower divergence at early stages.

timestep, we compute the average KL divergence between cross-attention maps corresponding to prompts with different object categories (e.g., "three dogs" vs. "four cats"), where each prompt is generated using a *different* initial noise seed. This group provides a reference scale for the magnitude of spatial divergence.

In all experiments, we evaluate ten object categories with target object counts ranging from 3 to 6 using the SDXL model. Figure 11 illustrates the temporal evolution of spatial layout divergence for Group 1 and Group 2. At $t = 0$, the KL divergence of Group 1 is approximately $8 \times 10^{-4}$, which is nearly 25 times smaller than that of Group 2 ($\approx 2 \times 10^{-2}$). As the denoising process progresses, the KL divergence of Group 1 increases gradually, whereas that of Group 2 grows much more rapidly, exhibiting a sharp rise after approximately five timesteps. These results demonstrate that early-stage spatial layouts are dominated by the initial noise, causing cross-attention patterns to remain highly stable across varying target object counts and thereby limiting the model's ability to satisfy numerical constraints.

### C.3. Empirical Evidence of the Noise-Induced Layout Prior

The existence of the *Noise-Induced Layout Prior* is supported by the following empirical observations:

- **Controlled visual analysis.** The controlled experiments in Figures 1 and 8 show that fixing the initial noise produces highly similar spatial layouts and early-stage cross-attention patterns, even when the target object count is changed.

- **Quantitative KL-divergence analysis.** The KL-

divergence results in Figure 11 and Appendix C.2 further quantify this phenomenon, showing substantially lower attention divergence under fixed-noise settings than under different-noise settings.

- **Extensive ablation studies.** The ablation results in Sec. 4.4 and Table 4 demonstrate that removing either the noise adjustment module or the early-attention regularization consistently degrades counting performance. These findings indicate that the *Noise-Induced Layout Prior* is a reproducible empirical phenomenon and a key factor underlying object-counting failures in diffusion models.

## D. Additional Experimental Results

### D.1. Additional Qualitative Analysis

**Qualitative Evaluation.** Figures 12, 13, and 14 present additional qualitative comparisons on the **COCOCount**, **T2I-Count**, and **MultiCount** benchmarks, respectively.

These results align with the findings presented in the main paper. Across all benchmarks, for both single-category and multi-category object counting tasks, our method consistently generates the precise number of objects specified in the prompt while maintaining high visual fidelity, significantly outperforming existing baselines.

Among diffusion-based text-to-image models, FLUX.1 exhibits relatively stronger counting accuracy when the target object count is small compared to other diffusion-based models such as Playground v2.5 and PixArt-Σ. However, as the target object count increases and in multi-category scenarios, FLUX.1 frequently exhibits under-generation or over-generation. In terms of visual quality, Playground v2.5 generally produces more aesthetically pleasing images.

Among object count control methods, CountGen delivers higher image quality than other counting-specific approaches. In contrast, Attention Refocus produces the lowest-quality images among the evaluated methods. This limitation stems from its reliance on bounding boxes generated by large language models, which often lack spatial diversity and result in overly regular object arrangements.

**Robustness to Prompt Complexity.** The prompts in our benchmark typically follow a simplified template, such as "A photo of $N$ [objects]". To evaluate the robustness of our method against more linguistically diverse instructions, we extend the original prompts with additional descriptive complexity. Figure 15 demonstrates that our method remains stable even as prompt complexity increases. Notably, counting accuracy is preserved while maintaining alignment with descriptive details such as background elements, confirming that our method handles dense semantic constraints effectively.

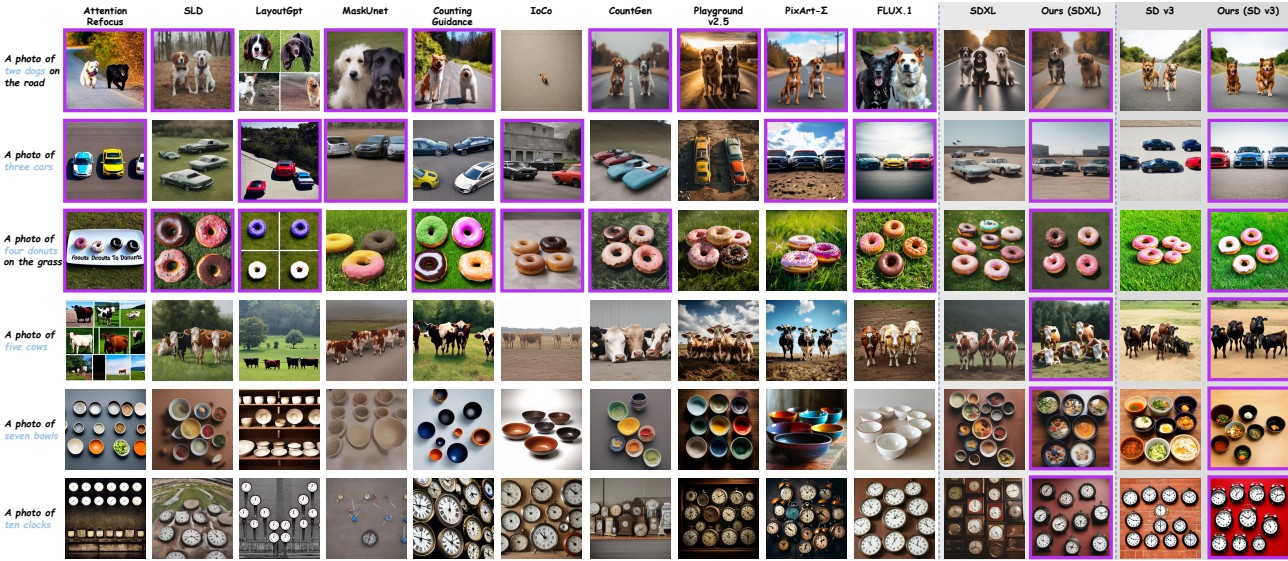

*Figure 12.* Qualitative comparison of baseline methods and our approach on the **COCOCount** benchmark. Images outlined in purple contain the correct number of objects.

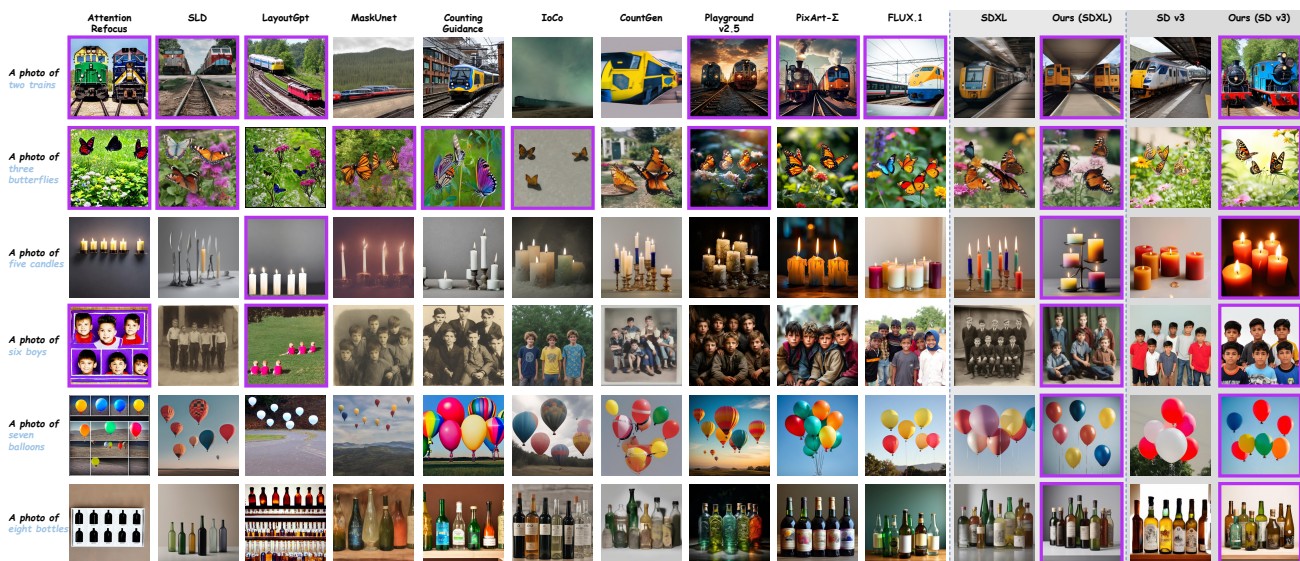

*Figure 13.* Qualitative comparison of baseline methods and our approach on the **T2I-Count** benchmark. Images outlined in purple contain the correct number of objects.

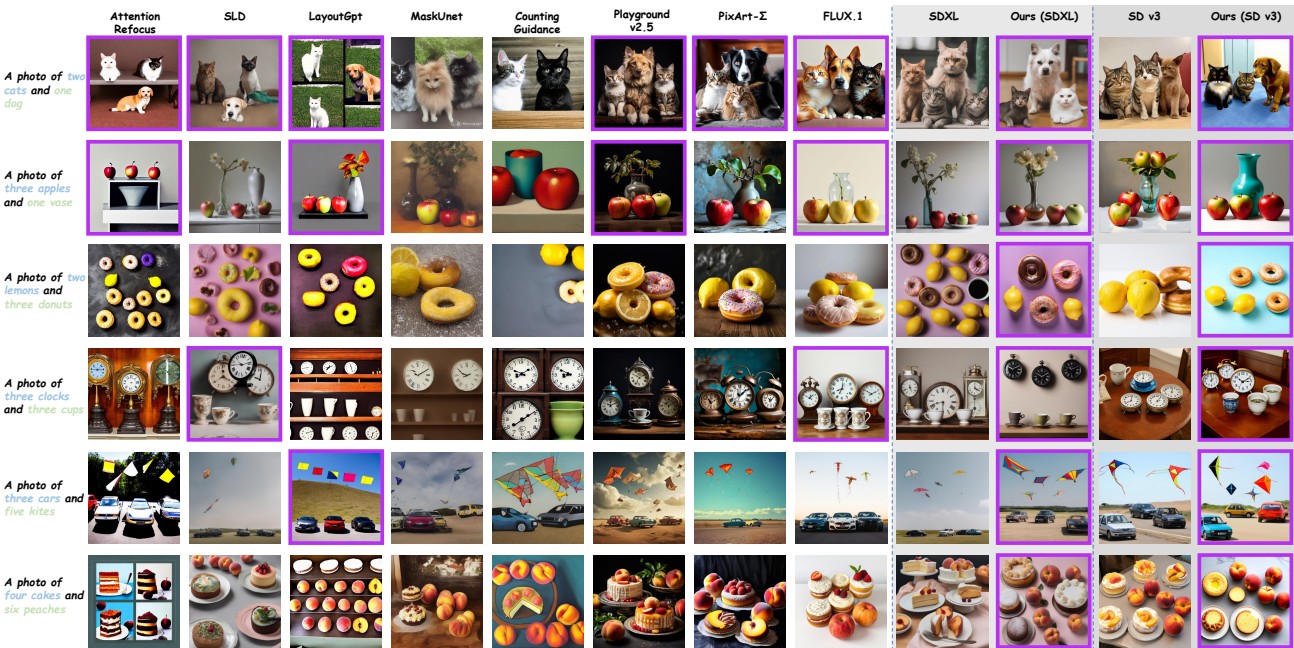

*Figure 14.* Qualitative comparison of baseline methods and our approach on the **MultiCount** benchmark. Images outlined in purple contain the correct number of objects.

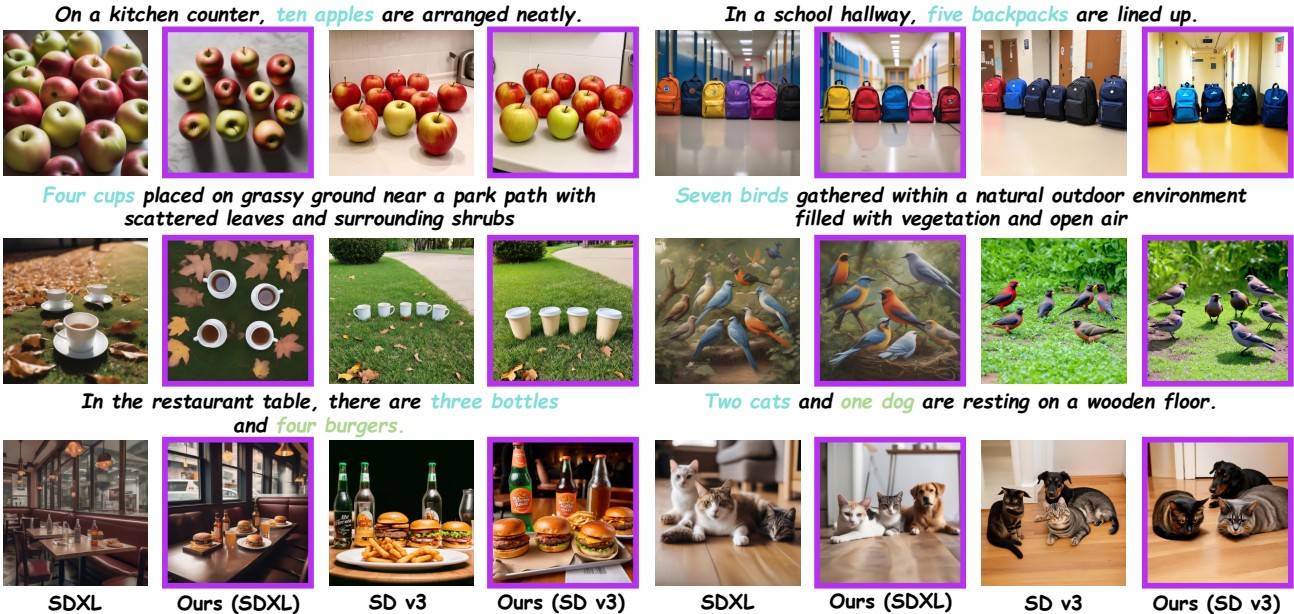

*Figure 15.* Counting stability under varied prompt formulations. Images outlined in purple contain the correct number of objects.

*Table 5.* Quantitative results on the **COCOCount**, **T2I-Count**, and **MultiCount** benchmarks.

| Method | STA↑ | Acc↑ | MAE↓ | IR↑ |
|---|---|---|---|---|
| *Quantitative results on the **COCOCount** benchmark.* | | | | |
| *w/o **Counting-Based Gate**.* | | | | |
| Ours (SDXL) | 0.8528 | 0.6125 | 0.9575 | 1.0414 |
| Ours (SD v3) | 0.8920 | 0.6225 | 0.6392 | 1.2681 |
| *w/ **Counting-Based Gate**.* | | | | |
| Ours (SDXL) | 0.8895 (+0.0367) | 0.6958 (+0.0833) | 0.7867 (-0.1708) | 1.0669 (+0.0255) |
| Ours (SD v3) | **0.9328** (+0.0408) | **0.7425** (+0.1200) | **0.4467** (-0.1925) | **1.2831** (+0.0150) |
| *Quantitative results on the **T2I-Count** benchmark.* | | | | |
| *w/o **Counting-Based Gate**.* | | | | |
| Ours (SDXL) | 0.8123 | 0.5556 | 0.9889 | 0.7248 |
| Ours (SD v3) | 0.8577 | 0.6022 | 0.7644 | **0.9355** |
| *w/ **Counting-Based Gate**.* | | | | |
| Ours (SDXL) | 0.8475 (+0.0352) | 0.6370 (+0.0814) | 0.8263 (-0.1626) | 0.7371 (+0.0123) |
| Ours (SD v3) | **0.8956** (+0.0379) | **0.7044** (+0.1022) | **0.5867** (-0.1777) | 0.9332 (-0.0023) |
| *Quantitative results on the **MultiCount** benchmark.* | | | | |
| *w/o **Counting-Based Gate**.* | | | | |
| Ours (SDXL) | 0.7060 | 0.2542 | 1.0965 | 1.3620 |
| Ours (SD v3) | 0.7903 | 0.3500 | 0.7514 | 1.4945 |
| *w/ **Counting-Based Gate**.* | | | | |
| Ours (SDXL) | 0.7190 (+0.0130) | 0.2917 (+0.0375) | 1.0569 (-0.0396) | 1.3650 (+0.0030) |
| Ours (SD v3) | **0.8061** (+0.0158) | **0.4083** (+0.0583) | **0.7076** (-0.0438) | **1.5036** (+0.0091) |

*Table 6.* Time and memory consumption. S-Time and M-Time correspond to the single-category and multi-category settings, respectively.

| Model | S-Time(s)↓ | M-Time(s)↓ | GPU(G)↓ |
|---|---|---|---|
| SDXL | 8.0 | 8.0 | 16 |
| SD v3 | 10.0 | 10.0 | 18 |
| FLUX.1 | 14.0 | 14.0 | 33 |
| PixArt-Σ | 2.9 | 2.9 | 15 |
| PlayGround v2.5 | 3.0 | 3.0 | 11 |
| LayoutGpt | 10.6 | 10.8 | 14 |
| **MaskUnet** | 61.9 | 61.9 | 18 |
| **SLD** | 49.8 | 64.1 | 28 |
| **Counting Guidance** | 13.4 | 15.7 | 15 |
| **IoCo** | 117.0 | – | 26 |
| CountGen | 42.0 | – | 50 |
| **Ours (SDXL)** | 36.0 | 85.6 | 23 |
| **Ours (SD v3)** | 46.0 | 110.1 | 50 |

### D.2. Additional Quantitative Analysis

**Impact of Initial Attention-Target Alignment.** Through extensive qualitative analysis, we observed that although our method is robust in most scenarios, redundant interventions can occasionally introduce unnecessary perturbations that lead to counting errors. This suggests that intervention is not always beneficial when the initial layout is already favorable. Motivated by this observation, we refine our framework by introducing **Counting-Based Gate**: if the base model already satisfies the target count, we directly adopt the output of the base model. As shown in Table 5, this adaptive strategy avoids over-correction in successful cases while preserving full intervention when necessary, leading to consistent improvements across all benchmarks.

**Computational Details.** We randomly sample 90 and 120 prompts with varying counts from COCOCount and MultiCount to evaluate different methods in single- and multi-category settings, respectively. All evaluations are performed on an A800 GPU, and Table 6 reports the per-image generation time and memory usage. Overall, all training-free methods (highlighted in bold) incur higher time and memory costs than conventional text-to-image models. In the single-category setting, our method still maintains a time advantage over other training-free baselines. However, in the multi-category setting, we assign one latent to each category and perform loss guidance sequentially within the same timestep, which approximately doubles the runtime. This limitation could be alleviated in future work through parallel computation.

**Quantitative results on T2I-Count.** Table 7 presents fine-grained quantitative comparisons for *single-category object counting* on the T2I-Count benchmark, complementing the main paper by reporting the previously omitted Precision, Recall, and F1 metrics. Across different base models, our method consistently improves both Precision and Recall, leading to superior F1 scores.

**Quantitative results on MultiCount.** Table 8 reports the

*Table 7.* Quantitative results on the **T2I-Count** benchmark.

| Method | STA↑ | Acc↑ | MAE↓ | Precision↑ | Recall↑ | F1↑ | IR↑ |
|---|---|---|---|---|---|---|---|
| FLUX.1 (Labs, 2024) | 0.7567 | 0.4089 | 1.2178 | 0.8882 | 0.9251 | 0.8830 | 0.8828 |
| PixArt-Σ (Chen et al., 2024) | 0.6945 | 0.3178 | 1.5956 | 0.8867 | 0.8770 | 0.8478 | 0.9672 |
| Playground v2.5 (Li et al., 2024b) | 0.7191 | 0.3778 | 1.5444 | 0.8896 | 0.8933 | 0.8584 | **1.0089** |
| Attention Refocus (Phung et al., 2024) | 0.6712 | 0.4489 | 2.5422 | 0.8174 | 0.9227 | 0.8296 | 0.0628 |
| SLD (Wu et al., 2024) | 0.6398 | 0.3333 | 1.9733 | 0.8319 | 0.7976 | 0.7703 | 0.0805 |
| LayoutGpt (Feng et al., 2023) | 0.6617 | 0.4756 | 3.1022 | 0.8008 | 0.9415 | 0.8272 | -0.2275 |
| MskUnet (Wang et al., 2025) | 0.6148 | 0.2733 | 2.3089 | 0.8613 | 0.8330 | 0.7928 | 0.9549 |
| Counting Guidance (Kang et al., 2025) | 0.6094 | 0.2467 | 2.1778 | 0.8668 | 0.8058 | 0.7816 | 0.1905 |
| IoCo (Zafar et al., 2026) | 0.7739 | 0.3622 | 1.3689 | 0.8804 | 0.8933 | 0.8594 | 0.4509 |
| CountGen (Binyamin et al., 2025) | 0.6879 | 0.4000 | 1.8533 | 0.8648 | 0.8960 | 0.8398 | 0.5059 |
| SDXL (Podell et al., 2024) | 0.6226 | 0.2978 | 2.2756 | 0.7944 | 0.9419 | 0.8272 | 0.6708 |
| SD v3 (Esser et al., 2024) | 0.7821 | 0.4444 | 1.1178 | 0.9027 | 0.9292 | 0.8941 | 0.8845 |
| Ours (SDXL) | 0.8123 | 0.5556 | 0.9889 | 0.9102 | **0.9512** | 0.9124 | 0.7248 |
| Ours (SD v3) | **0.8577** | **0.6022** | **0.7644** | **0.9435** | 0.9410 | **0.9271** | 0.9355 |

*Table 8.* Quantitative results on the **MultiCount** benchmark.

| Method | STA↑ | Acc↑ | MAE↓ | Precision↑ | Recall↑ | F1↑ | IR↑ |
|---|---|---|---|---|---|---|---|
| FLUX.1 (Labs, 2024) | 0.7441 | 0.2292 | 0.9347 | 0.9088 | 0.8544 | 0.8523 | 1.4826 |
| PixArt-Σ (Chen et al., 2024) | 0.5563 | 0.0722 | 1.6132 | 0.8008 | 0.6656 | 0.6825 | 1.3353 |
| PlayGround v2.5 (Li et al., 2024b) | 0.6245 | 0.1222 | 1.4028 | 0.8548 | 0.7808 | 0.7691 | 1.4480 |
| Attention Refocus (Phung et al., 2024) | 0.4685 | 0.1028 | 1.9486 | 0.6561 | 0.5178 | 0.5434 | -0.0678 |
| SLD (Wu et al., 2024) | 0.4980 | 0.0986 | 1.8799 | 0.7314 | 0.5337 | 0.5788 | 0.9665 |
| LayoutGpt (Feng et al., 2023) | 0.4619 | 0.1125 | 2.0500 | 0.6158 | 0.5331 | 0.5370 | 0.0459 |
| MskUnet (Wang et al., 2025) | 0.3885 | 0.0250 | 2.2062 | 0.6879 | 0.4338 | 0.4886 | 1.3425 |
| Counting Guidance (Kang et al., 2025) | 0.3694 | 0.0194 | 2.2299 | 0.6666 | 0.4287 | 0.4763 | 0.3603 |
| SDXL (Podell et al., 2024) | 0.5550 | 0.0833 | 1.8882 | 0.7781 | 0.8145 | 0.7394 | 1.2096 |
| SD v3 (Esser et al., 2024) | 0.6569 | 0.1486 | 1.2236 | 0.8685 | 0.7796 | 0.7823 | 1.4238 |
| Ours (SDXL) | 0.7060 | 0.2542 | 1.0965 | 0.8721 | **0.8833** | 0.8410 | 1.3620 |
| Ours (SD v3) | **0.7903** | **0.3500** | **0.7514** | **0.9278** | 0.8696 | **0.8747** | **1.4945** |

corresponding quantitative results for *multi-category object counting* on the MultiCount benchmark, likewise supplementing the main paper with Precision, Recall, and F1 metrics. Our approach achieves consistent gains in Precision and Recall across all evaluated base models, resulting in improved F1 performance. For prompts containing multiple object categories, all metrics are computed independently for each category and then averaged across categories to obtain the final score, ensuring a balanced evaluation of multi-category object counting performance.

### D.3. Additional Ablation Studies

**Ablation Study on MultiCount.** We conduct ablation studies on the MultiCount benchmark to assess the contribution of each component in our method. Specifically, we evaluate the following variants: (1) w/o OSE: removing *Object Saliency Enhancement*; (2) w/o LRR + OSE: removing the *Count-Aware Noise Adjustment Strategy*; (3) w/o $\mathcal{L}_{acl}$: removing the *Attention Concentration Loss*; (4) w/o $\mathcal{L}_{drl}$: removing the *Dispersion Regularization Loss*; (5) w/o $\mathcal{L}_{ssl}$: removing the *Semantic Suppression Loss*; (6) w/o $\mathcal{L}_{acl}$ +

*Table 9.* Quantitative ablation results on the **MultiCount** benchmark.

| Ablation | STA↑ | Acc↑ | MAE↓ | IR↑ |
|---|---|---|---|---|
| w/o OSE | 0.6912 | 0.2153 | 1.1674 | 1.3409 |
| w/o LRR + OSE | 0.6726 | 0.2111 | 1.2563 | 1.3243 |
| w/o $\mathcal{L}_{acl}$ | 0.5925 | 0.1042 | 1.6229 | 1.2681 |
| w/o $\mathcal{L}_{drl}$ | 0.6679 | 0.2069 | 1.2292 | 1.2302 |
| w/o $\mathcal{L}_{ssl}$ | 0.6793 | 0.2111 | 1.1792 | **1.3665** |
| w/o $\mathcal{L}_{acl}$ + $\mathcal{L}_{drl}$ + $\mathcal{L}_{ssl}$ | 0.5584 | 0.0847 | 1.7042 | 1.2381 |
| Ours (SDXL) | **0.7060** | **0.2542** | **1.0965** | 1.3620 |

$\mathcal{L}_{drl}$ + $\mathcal{L}_{ssl}$: removing the entire *Attention-Guided Layout Consistency Strategy*.

Table 9 summarizes the quantitative results. Consistent with observations in the main paper, removing any individual component leads to a clear degradation in counting performance, highlighting the importance of each design choice in multi-category object counting. Several key insights can be drawn:

*Table 10.* Effect of region geometry on the **COCOCount**, **CompBench**, and **MultiCount** benchmarks.

| Method | COCOCount | | | CompBench | | | MultiCount | | |
|---|---|---|---|---|---|---|---|---|---|
| | STA↑ | Acc↑ | IR↑ | STA↑ | Acc↑ | IR↑ | STA↑ | Acc↑ | IR↑ |
| Square | **0.8596** | **0.6205** | 1.0045 | 0.8076 | **0.5644** | 0.7188 | 0.7033 | 0.2403 | 1.2991 |
| Triangular | 0.8423 | 0.5933 | 1.0230 | 0.7899 | 0.5578 | 0.6992 | 0.6906 | 0.2306 | 1.3315 |
| Circular (Ours (SDXL)) | 0.8528 | 0.6125 | **1.0414** | **0.8123** | 0.5556 | **0.7248** | **0.7060** | **0.2542** | **1.3620** |

- **Importance of Structured Spatial Guidance.** The most severe performance degradation occurs when the attention-guided layout consistency strategy is entirely removed, indicating that structured spatial guidance is central to robust multi-category object counting.

- **Effect of Individual Loss Terms.** Among the proposed losses, the *Attention Concentration Loss* $\mathcal{L}_{acl}$ has the largest impact. Its removal leads to substantial drops in STA and Accuracy, underscoring its role in preventing ambiguous object instances.

- **Trade-off Between Counting Accuracy and Image Quality.** Removing the *Semantic Suppression Loss* (w/o $\mathcal{L}_{ssl}$) results in reduced counting accuracy while yielding the highest ImageReward score. This indicates that, although image quality may improve slightly without semantic suppression, the absence of this loss leads to increased object leakage across categories, harming precise multi-category object counting.

**Impact of Region Geometry.** In the *Object Region Planning Algorithm*, the concentrated mask $M_c^{(i)}$ is defined as a circular area centered at the region centroid $c^{(i)}$. To assess the sensitivity of our method to this geometric choice, we conduct an ablation study replacing the default circular mask with square and triangular geometries while maintaining all other hyperparameter settings. The results, summarized in Table 10, indicate that the circular region is the most effective configuration. Specifically, for the COCOCount and T2I-Count benchmarks, the circular geometry achieves the highest IR score, with only negligible variations observed across other counting metrics. Furthermore, on the more challenging MultiCount benchmark, the circular mask consistently outperforms the alternative shapes across all evaluation metrics. This suggests that the inherent symmetry of a circle better aligns with the isotropic nature of attention diffusion, providing more stable spatial priors for object generation.

**Impact of Random Seed Count.** We further investigate the empirical stability of our SDXL-based method by conducting a sensitivity analysis on the number of random seeds used for evaluation on the T2I-Count benchmark. For each prompt, images are generated using a fixed set of randomly sampled seeds. As illustrated in Figure 16, as the number

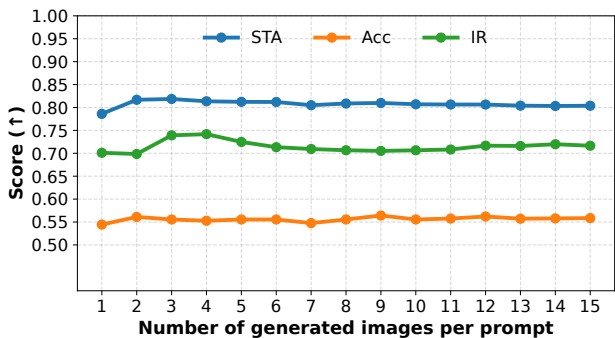

*Figure 16.* The impact of random seed on generation accuracy and visual quality.

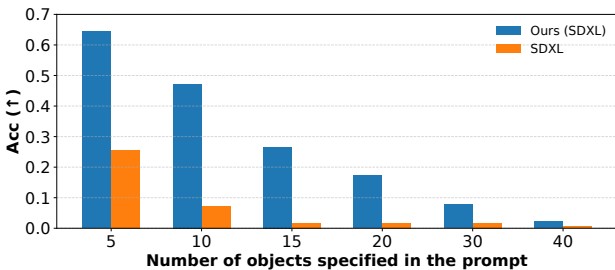

*Figure 17.* The impact of object count on generation accuracy.

of seeds increases, all evaluation metrics rapidly converge and exhibit only negligible fluctuations beyond a certain threshold. These results indicate that a modest number of seeds is sufficient to obtain statistically reliable performance estimates. To strike an optimal balance between computational efficiency and reporting confidence, we utilize five random seeds for all quantitative experiments in this study.

**Robustness to High Object Counts.** We further evaluate the robustness of our SDXL-based method as the target object count increases. Specifically, we select 18 object categories from the COCOCount benchmark and construct simplified prompts following the format "a photo of $N$ apples", where $N \in \{5, 10, 15, 20, 30, 40\}$. For each prompt, we generate ten images using ten randomly sampled seeds. To ensure fair comparisons, the same set of seeds is used across all prompts. Figure 17 compares the accuracy of our method with the base SDXL model as the target object count increases. As the number of objects grows, the performance of SDXL degrades rapidly. For instance, when

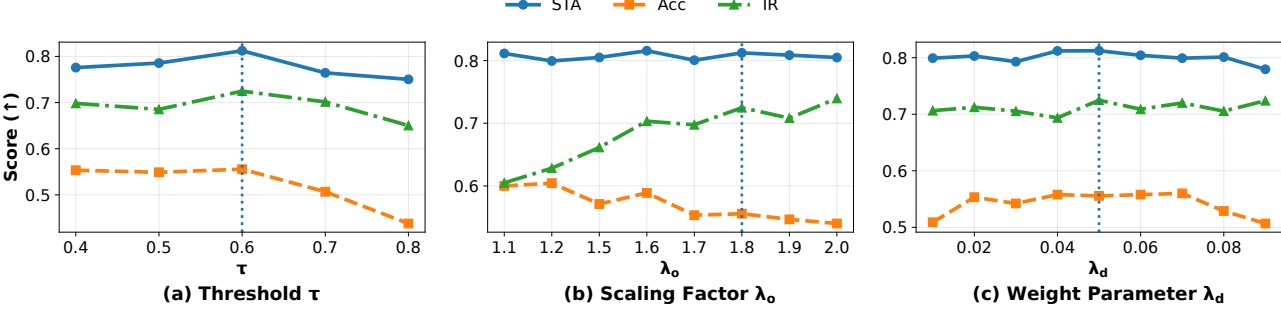

*Figure 18.* Ablation of three key hyperparameters for single-category object counting on the T2I-Count benchmark. (a) Effect of the attention threshold $\tau$. (b) Effect of the scaling factor $\lambda_o$. (c) Effect of the weighting parameter $\lambda_d$.

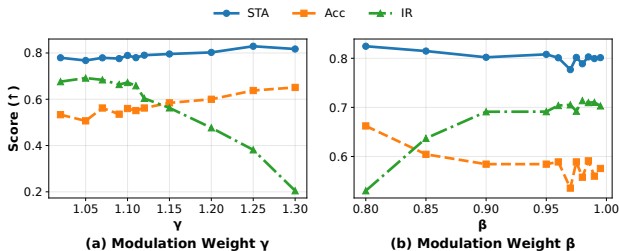

*Figure 19.* Sensitivity analysis of Object Saliency Enhancement hyperparameters. (a) Effect of the amplification weight $\gamma$ on counting accuracy and visual quality, with $\beta$ fixed to 1.0. (b) Effect of the background suppression weight $\beta$ on counting accuracy and visual quality, with $\gamma$ fixed to 1.0.

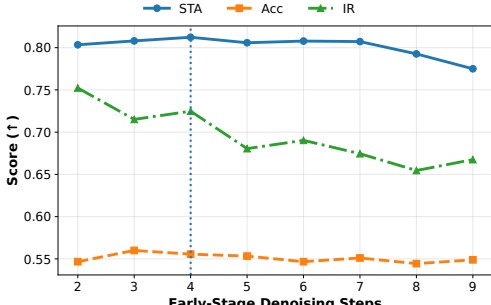

*Figure 20.* Effect of early-stage denoising steps in the Attention-Guided Layout Consistency Strategy for single-category object counting on the T2I-Count benchmark.

the object count reaches 15, SDXL achieves an accuracy of less than 0.02, whereas our method attains a substantially higher accuracy of approximately 2.27. Nevertheless, as the object count further increases to 30 or even 40, the accuracy of our method also drops below 0.1. These results suggest that accurately modeling and controlling scenes with a large number of objects remains a significant challenge for current text-to-image generation models.

### D.4. Hyperparameter Analysis

We conduct experiments on key hyperparameters of our SDXL-based method on the T2I-Count benchmark to analyze their impact on model performance.

Specifically, we first examine three key hyperparameters: the attention threshold $\tau$, the scaling factor $\lambda_o$, and the weighting parameter $\lambda_d$. The attention threshold $\tau$ plays a crucial role in ensuring stable region partitioning in the *Object Region Planning*. As shown in Figure 18(a), both counting accuracy and visual quality achieve their highest values when $\tau = 0.6$. The scaling factor $\lambda_o$ controls the selection of highly activated regions in the *Attention-Guided Layout Consistency Strategy*. As illustrated in Figure 18(b), setting $\lambda_o = 1.8$ provides the best trade-off between counting accuracy and visual quality. The weighting parameter

$\lambda_d$ balances the positive and negative components of the *Attention Concentration Loss*. As shown in Figure 18(c), optimal performance in both counting accuracy and image quality is achieved when $\lambda_d = 0.05$.

We further investigate two modulation weights, $\gamma$ and $\beta$, in the *Object Saliency Enhancement* module, which respectively control the amplification of object-relevant regions and the mild suppression of background activations. To identify suitable values, we adopt a two-stage search strategy. First, we fix $\beta = 1.0$ and vary $\gamma$ over a broad range, $\gamma \in [1.0, 1.3]$, to examine its influence on overall performance. As shown in Figure 19(a), $\gamma = 1.1$ yields the best trade-off between counting accuracy and visual quality. Next, we analyze the effect of $\beta$ by varying it within $[0.8, 1.0]$, as illustrated in Figure 19(b). When $\beta < 0.9$, image quality degrades substantially, indicating that excessive background suppression disrupts the model's original generation capability. In contrast, image quality remains stable or even improves when $\beta$ lies in a higher range. Based on this observation, we further perform a fine-grained grid search over $\beta \in [0.95, 0.99]$ with a step size of 0.005, while fixing $\gamma = 1.1$. Finally, $\beta = 0.975$ achieves the best overall performance in terms of both counting accuracy and visual quality.

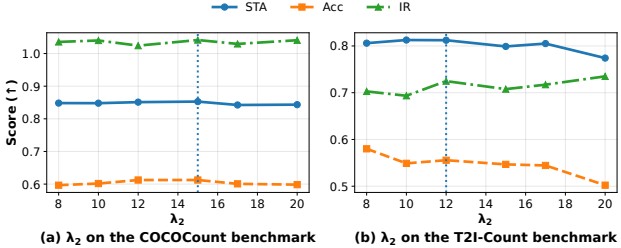

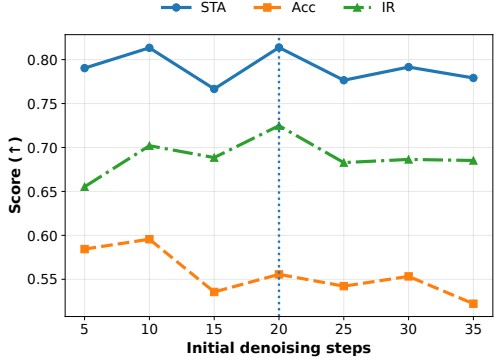

*Figure 21.* Ablation of $\lambda_2$ on the COCOCount and T2I-Count benchmarks. (a) Effect of $\lambda_2$ on the COCOCount benchmark. (b) Effect of $\lambda_2$ on the T2I-Count benchmark.

*Figure 23.* Effect of initial denoising steps $S$ on the T2I-Count benchmark.

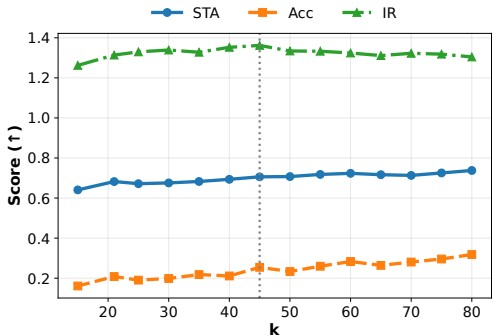

*Figure 22.* Effect of Top-k strategy for multi-category object counting on the MultiCount benchmark.

Moreover, we examine the influence of the initial denoising stages within the *Attention-Guided Layout Consistency Strategy*. As shown in Figure 20, performing optimization over the first four denoising steps yields the best performance in terms of both counting accuracy and visual quality.

We further evaluate the impact of loss weights on the CO-COCount and T2I-Count benchmarks. Specifically, we fix $\lambda_1 = 1$ and vary $\lambda_2$. As shown in Figure 21, $\lambda_2 = 12$ and $\lambda_2 = 15$ provide the best trade-off between counting accuracy and visual quality on COCOCount and T2I-Count, respectively.

Additionally, we conduct an ablation study on the Top-k% strategy for multi-category object counting on the Multi-Count benchmark. As shown in Figures 22, too small $k$ leads to insufficient object coverage, while overly large values introduce noisy regions and degrade the image quality; a moderate setting (e.g., $k = 45$) achieves the best trade-off.

Finally, we analyze the sensitivity to the number of early sampling steps $S$ within the *Count-Aware Noise Adjustment Strategy* on the T2I-Count benchmark. As illustrated in Figures 23, the results indicate that the parameters are relatively insensitive, and that $S = 20$ achieves the best trade-off between counting accuracy and visual quality.

The analyses presented in Figures 18–23 reveal that performance is primarily governed by only two key parameters, $\tau$

and $\lambda_o$. Most other hyperparameters remain stable across a wide range and can therefore be fixed at their default values. This concentrated sensitivity substantially simplifies the tuning process and makes the method practical for diverse generation tasks.

## E. User Study

### E.1. Human Evaluation for the STA Metric.

We conduct a human study on 120 generated images from the COCOCount benchmark, and evaluate the Pearson, Spearman, and Kendall correlations between automatic metrics and human judgments. As shown in Table 11, both STA and F1-score exhibit substantially better correlations with human evaluations than conventional metrics. Compared with Acc, STA shows consistently higher correlation (e.g., Pearson: 0.78 vs. 0.60), highlighting that Acc is overly strict as it only considers exact matches and ignores near-correct results. In contrast, STA is a scale-aware metric that captures relative counting accuracy with bounded penalties. For example, predicting 9 objects for a target of 10 yields STA = 0.90, whereas predicting 1 object for a target of 2 gives STA = 0.50, reflecting the intuition that missing one object out of two is much more severe than missing one out of ten. In contrast, F1 assigns 0.95 and 0.67 to these two cases, respectively, compressing this perceptual gap. This difference becomes more pronounced under severe over-counting: when the target is 2 and the prediction is 8, STA drops to 0, while F1 remains 0.40. These examples indicate that STA more faithfully captures the relative severity of counting errors. Therefore, STA is more interpretable and scale-aware, making it more consistent with human judgment.

### E.2. Human Evaluation of the Method.

We conducted two user studies, each involving ten participants, to evaluate quantitative counting accuracy and qualitative perceptual quality, respectively. Both studies used

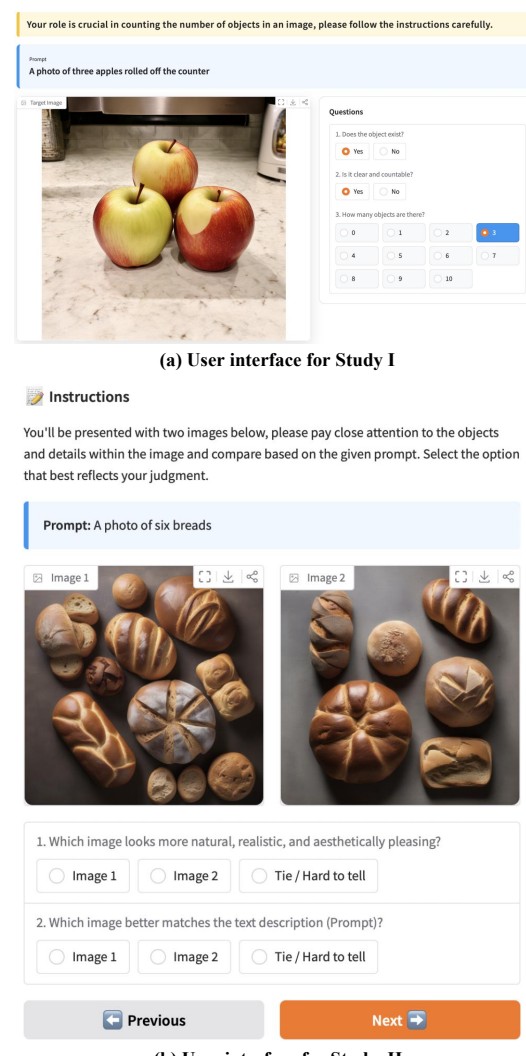

(a) User interface for Study I

(b) User interface for Study II

*Figure 24.* User interfaces for two user studies.

*Table 11.* Correlation analysis between automatic metrics and human evaluation

| Metric | Person | Spearman | Kendall |
|---|---|---|---|
| STA | 0.78 | 0.77 | 0.70 |
| F1 | **0.83** | **0.78** | **0.71** |
| Acc | 0.60 | 0.72 | 0.66 |
| Precision | 0.62 | 0.51 | 0.46 |
| Recall | 0.42 | 0.45 | 0.43 |

are required to select the exact number of observed objects.

The quantitative results of our user study are summarized in Figure 25, illustrating the performance of our method for both single-category and multi-category object counting tasks. As shown, the human-evaluated accuracy closely aligns with the automated calculated accuracy metric, indicating strong consistency between human judgments and quantitative evaluation. In the *single-category* setting, our method substantially improves the counting precision of the SDXL and SD v3 base models, achieving performance that surpasses even advanced models such as FLUX.1. This advantage becomes more pronounced in the *multi-category* setting, where increased prompt complexity poses greater challenges for numerical control. Notably, our approach maintains robust counting accuracy under these conditions. The strong correlation between human evaluations and automated metrics not only validates the reliability of our evaluation protocol but also demonstrates that the proposed *Count-Aware Noise Adjustment* and *Attention-Guided Layout Consistency* strategies yield perceptible and meaningful improvements in object counting performance from an end-user perspective.

**Study II: Qualitative Perceptual Evaluation.** The second study is designed to assess whether our method degrades the inherent generative capabilities of the base models (i.e., SDXL and SD v3). For each prompt, an image is generated either by our method or by the corresponding base model. Figure 24(b) presents the evaluation interface, which shows a textual prompt along with the generated image. Participants are asked to evaluate the images in terms of visual quality and image–text alignment.

The results, shown in Figure 26, reveal two key findings. First, in terms of *Visual Quality*, the high proportion of "Tie" votes indicates that images generated by our method are perceptually indistinguishable from the high-fidelity outputs of the original SDXL and SD v3 base models. This confirms that our approach preserves the aesthetic quality of the underlying models. Second, our method exhibits a clear advantage in *Image–Text Alignment* across both single-category and multi-category object counting tasks. By alleviating noise-induced layout priors, our approach improves counting accuracy while maintaining strong visual fidelity.

a total of 210 test prompts, including 90 single-category prompts from the T2I-Count benchmark and 120 multi-category prompts from the MultiCount benchmark.

**Study I: Quantitative Counting Accuracy.** The first study evaluates the model's adherence to numerical instructions. In this user study, we compare our method against four representative baselines (i.e., SDXL, SD v3, Flux, and Count-Gen) for *single-category object counting*, and against three baselines (i.e., SDXL, SD v3, and Flux) for multi-category object counting. For each prompt, a single image is generated by either our method or one of the baseline approaches. Figure 24(a) illustrates the user interface, which displays a textual prompt together with a generated image. Participants are first asked to verify the presence and countability of the target objects. If the generated objects are deemed uncountable, the count is recorded as zero. Otherwise, participants

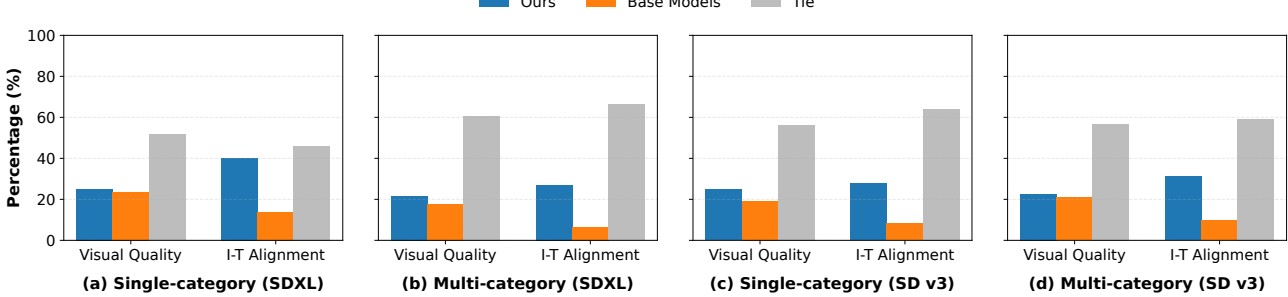

*Figure 25.* Quantitative results of the user study comparing human-evaluated accuracy and automated counting accuracy for single-category and multi-category object counting tasks.

*Figure 26.* Qualitative comparison of visual quality and image–text alignment between our method and the SDXL/SD v3 base models for both single-category and multi-category object counting tasks. Bars indicate user preference rates for *Visual Quality* and *Image–Text Alignment*.

