# OpenReview forum: "Mitigating Noise-Induced Layout Priors for Object Counting in Diffusion Models"
_ICML.cc/2026/Conference — ICML 2026 regular_

### Official Review · Reviewer_yy9U · 2026-02-23

**Soundness:** 2
**Presentation:** 3
**Significance:** 2
**Originality:** 2
**Overall Recommendation:** 4
**Confidence:** 5

**Summary:**

This paper argues that initial latent noise induces a strong prior over early cross-attention patterns in text-to-image diffusion models, which in turn fixes spatial layouts and often dominates object-count prompts. To mitigate this “Noise-Induced Layout Prior,” the authors propose a training-free inference-time framework. Experiments on COCOCount, T2I-Count, and a newly introduced MultiCount benchmark report improved counting accuracy with relatively preserved image quality compared to several diffusion backbones and prior count-control baselines.

**Compliance With Llm Reviewing Policy:**

Affirmed.

**Key Questions For Authors:**

Please refer to the weaknesses.

**Limitations:**

yes

**Strengths And Weaknesses:**

# Strengths
* Clear empirical motivation that initial noise can dominate count control, supported by multiple visualizations, notably Figure 1(a,c) and Figure 7, which show near-invariant layouts under fixed noise despite changing count prompts.
* The method is training-free and model-agnostic in principle, demonstrated on both UNet-based SDXL and transformer-based SD v3.
* Helpful intermediate visualizations that make the mechanism less “black box,” particularly Figure 5 (attention map → clusters → Voronoi regions → final image), which concretely illustrates how planning aligns with final object placements.
---
# Weaknesses
* The proposed method requires (i) an initial (S)-step planning run and (ii) gradient-based latent updates for early steps (Eq. 15). None of the tables report wall-clock time, GPU hours, or number of additional forward/backward passes. Since “training-free” methods often trade training cost for test-time cost, this omission matters for practical adoption and fair comparison to baselines like IoCo or Counting Guidance that also do test-time optimization.
* Region planning relies on attention clustering into exactly (K) instances, which can be brittle. The paper shows clean examples in Figure 5, but does not quantify how often planning fails, how sensitive it is to (S), $\tau$, or prompt tokenization, and what happens downstream when clusters are wrong.
* MultiCount benchmark is under-specified in the main body. It is introduced as “new,” but lacks enough detail (example prompts, difficulty characterization) to assess whether the gains generalize beyond templated scenarios.
* Literature positioning is incomplete for closely related noise-layout and training-free counting-control work [1,2,3,4,5].


[1] Dahary O, Cohen Y, Patashnik O, et al. Be decisive: Noise-induced layouts for multi-subject generation[C]//Proceedings of the Special Interest Group on Computer Graphics and Interactive Techniques Conference Conference Papers. 2025: 1-12.

[2] Mondal A, Banerjee A, Nag S, et al. CountLoop: Training-Free High-Instance Image Generation via Iterative Agent Guidance[J]. arXiv preprint arXiv:2508.16644, 2025.

[3] Li Y, Wan P, Han L, et al. CountDiffusion: Text-to-Image Synthesis with Training-Free Counting-Guidance Diffusion[J]. arXiv preprint arXiv:2505.04347, 2025.

[4] Zafar O, Cohen Y, Wolf L, et al. Detection-Driven Object Count Optimization for Text-to-Image Diffusion Models[J]. arXiv preprint arXiv:2408.11721, 2024.

[5] Wang L, Li S, Yang F, et al. Not All Parameters Matter: Masking Diffusion Models for Enhancing Generation Ability[C]//Proceedings of the Computer Vision and Pattern Recognition Conference. 2025: 12880-12890.

---

> ### Author Rebuttal · Authors · 2026-03-31
>
> We appreciate all the constructive feedback from the reviewer.
>
>
> **Q1: Analysis of Computational Overhead.**
>
> **A1:** Please refer to our response to Reviewer mdPw’s first question.
>
> **Q2: Region Planning Analysis.**
>
> **A2:**
>
> 1. We visualize the region planning process but find it hard to directly assess failures in the final partitioning (represented by a Voronoi diagram). Nevertheless, this stage is critical, as it affects both the LRR and OSE modules and the Attention-Guided Layout Consistency Strategy. Therefore, we evaluate it indirectly via ablation studies on downstream modules and sensitivity analysis (see Tables 4 & 7 and Fig. 15(a)).
>
> 2. We further analyze K-means clustering, which aims to keep cluster centers well separated and located in high-response attention regions. Accordingly, we define two failure cases: (i) cluster centers are too close to distinguish their associated instances; (ii) a center falls in a low-response attention region, leading to imbalanced clusters and leaving downstream modules without reliable local anchors. Using 120 randomly sampled generated images from the T2I-Count benchmark, we manually identify K-means clustering failures. The results are as follows: (1) 3 cases of the first type, among which 1 still produces the correct count; (2) 7 cases of the second type, among which 3 still produce the correct count. Additionally, there is 1 case where both (1) and (2) occur, and the final result is still correct.
>
> 3. The above observation motivates us to design detection and refinement mechanisms for the two K-means failure cases. **Detection Mechanism**: (1) Use normalized distances between cluster centers to identify overly close centers. (2) Use the sum of normalized attention within a circular region around each center to detect low-response regions. **Refinement Mechanism**: If failure is detected, we relocate problematic centers to high-response regions while enforcing distance constraints.
>
> 4. We further analyze the sensitivity to the number of early sampling steps S under the same experimental setup as Sec. D.4. The results (Table 1) show that S=20 achieves the best trade-off between counting accuracy and visual quality. See Fig. 15(a) for the sensitivity analysis of the threshold $\tau$.
>
> **Table 1: sensitivity analysis.**
> | S  | STA  | Acc  | IR   |
> |----|------|------|------|
> | 5  | 0.79 | 0.58 | 0.66 |
> | 10 | **0.81** | **0.60** | 0.70 |
> | 15 | 0.77 | 0.54 | 0.69 |
> | 20 | **0.81** | 0.56 | **0.72** |
> | 30 | 0.79 | 0.55 | 0.69 |
>
>
> **Q3: MultiCount Benchmark Description.**
>
> **A3:** Following COCOCount, MultiCount uses prompts of the form “a photo of N1 object1 and N2 object2,” with N1,N2∈[1,9] and K∈[3,10], yielding 120 prompts. Categories are drawn from COCOCount and T2I-Count, with implausible pairs filtered via GPT-5. Compared to single-category settings, MultiCount is more challenging due to inter-category interference and joint control of multiple object groups. Details will be included in the appendix.
>
>
> **Q4: Comprehensive Literature.**
>
> **A4:** We have added the requested comparisons (detailed below), which will be incorporated into the revised manuscript.
>
> 1. Our IoCo baseline and [4] correspond to different versions of the same work. We do not include [1] for comparison, as its results are not directly comparable to our metrics.
>
> 2. Comparison with [2] and [3]. As no official code is available for CountDiffusion [3] and CountLoop [2], we follow their reported benchmark and evaluation protocol for fair comparison.
>
> Table 2 shows that our method achieves higher accuracy and stronger overall performance, especially with advanced base models (e.g., SD3), demonstrating better generalization.
>
> **Table 2: quantitative results.**
> | Model               | MAE    | Acc    | IR     |
> |-------------------------------|--------|--------|--------|
> | CountDiffusion (SDXL)      |0.90|0.59|1.02|
> | CountDiffusion (PixArt-$\Sigma$) |0.89| 0.60|1.21|
> | Ours (SDXL)               | 1.01| 0.65 | 1.07   |
> | Ours (SD3)                 | **0.62**  | **0.67** | **1.28** |
>
> As shown in Table 3, while CountLoop reports very low MAE, it only evaluates this single metric, which raises concerns about its reliability. Notably, our reproduced MAE matches CountDiffusion after adjusting the detection tool, but deviates significantly from CountLoop, indicating possible evaluation inconsistencies.
>
> **Table 3: quantitative results.**
> | Model   | COCOCount | T2I-Count |
> |-------------|-----------|-----------|
> | CountLoop   | **0.45** | **1.23**|
> | Ours (SDXL) |2.45| 3.16|
> | Ours (SD3)  |1.24| 2.09|
>
> 3. Comparison results with [5]. Using the official MaskUnet [5] code under our protocol, our method consistently outperforms it across all benchmarks (Table 4).
>
> **Table 4: quantitative results.**
> | Benchmark| STA| Acc|
> |------------|------|-----|
> | COCOCount|0.67|0.30|
> | T2I-Count| 0.61| 0.27|
> | MultiCount |0.39|0.03|
>
> **All quantitative results are rounded to two decimal places for brevity.**

---

> > ### Author Rebuttal · Reviewer_yy9U · 2026-04-01
> >
> > The authors’ rebuttal has addressed my concerns, and I am therefore raising my score to Weak Accept.

---

> > > ### Author Response · Authors · 2026-04-01
> > >
> > > Thank you very much for your positive feedback. We are truly encouraged to hear that our rebuttal has addressed your concerns. We will continue to refine and strengthen the work, and we greatly appreciate the opportunity for further engagement in advancing this research.

---

### Official Review · Reviewer_UAHz · 2026-03-09

**Soundness:** 3
**Presentation:** 3
**Significance:** 3
**Originality:** 3
**Overall Recommendation:** 4
**Confidence:** 4

**Summary:**

This paper investigates why text-to-image (T2I) diffusion models fail to generate the correct number of objects, attributing this to the explicit influence of initial noise on spatial layout formation—formulated as a "Noise-Induced Layout Prior." A training-free framework is proposed, comprising: (1) a Count-Aware Noise Adjustment Strategy, which manipulates the initial latent noise to align it with the count of the target objects; and (2) an Attention-Guided Layout Consistency Strategy, which performs test-time (inference-time) optimization on early cross-attention maps. Experiments on single-class and multi-class benchmarks demonstrate that the proposed approach outperforms baseline models.

**Compliance With Llm Reviewing Policy:**

Affirmed.

**Key Questions For Authors:**

1. How does the method perform when the initial attention map has already matched the target count? In this case, does noise adjustment introduce unnecessary perturbations?

2. What happens when K-means produces extremely imbalanced clusters? Is there a mechanism to detect and handle degenerate region assignments?

3. Can the authors provide evidence that the "noise-guided layout prior" applies to models with different architectures (e.g., DiT-based FLUX versus UNet-based SDXL)? Current experiments use SDXL and SD v3, but the observations in Figure 1 should be validated more broadly.

4. How sensitive is this method to the initial denoising steps $S$ used for region planning?

5. Can this method be generalized to uncountable compositional properties (e.g., spatial relationships, color)?

Please note that if some questions/weakness are not solved in rebuttal, I will lower my score.

**Limitations:**

yes

**Strengths And Weaknesses:**

S1. Core Observation with Clear Motivation

The insightful observation that initial noise has a dominant influence on object layout, and that early cross-attention plays a mediating role, is strongly supported by the controlled experiments in Figure 1. The demonstration that fixing the noise while changing the counting labels produces nearly identical layouts (Figure 1a) is particularly compelling and provides a clear, falsifiable observation that forms a strong motivation for the method.

S2. Clear and Modular Framework Design

The overall framework has a rigorous logical structure. Decomposing it into noise-level intervention (LRR + OSE) and attention-level optimization (ACL + DRL + SSL) is reasonable, and each module addresses a specific failure mode. Ablation experiments confirm that both components contribute substantially to the final performance.

S3. Training-Free and Model-Independent

The method requires no fine-tuning and has been demonstrated on two different base models (SDXL and SD v3) showing reasonable generalization ability. This is highly valuable in practical applications due to avoiding the cost and data requirements of retraining.

S4. Comprehensive Evaluation

The paper evaluates the model on three benchmarks (COCOCount, T2I-Count, MultiCount), compares it with nine baseline models (five generative models + four counting control methods), and reports several complementary metrics, including the newly proposed STA. Ablation experiments systematically separate the contribution of each module.

W1. Overstated Claim of "Noise-Guided Layout Prior"—Insufficient Formalization

The paper claims to "formalize" a noise-guided layout prior (Observation 1, Section 3.1), but this formalization is essentially a qualitative observation restated in mathematical notation. No formal definition of "spatial structure $S$" is given, no quantitative metric measures "highly similar spatial structures," and no statistical tests are provided to establish the strength or consistency of this prior across various noisy samples. This is a descriptive observation, not a rigorous formalization.

Provide rigorous quantitative analysis, such as measuring the similarity of cross-attention maps (e.g., SSIM, cosine similarity) for different cue words under fixed noise and comparing it to a null distribution. Report based on statistics from a large number of random seeds and cue words. Furthermore, discuss under what conditions this prior might weaken (e.g., completely different object categories, different model architectures).

W2. Object Region Planning Relies on Initial Denoising Transmission

The object region planning module runs $S$ steps of denoising to aggregate the cross-attention maps before applying noise adjustment. This means the method relies on the cross-attention patterns generated by the original (unadjusted) noise, which, according to the paper's own arguments, may have encoded incorrect layout priors. If the initial noise elicits (e.g.) two object clusters, and the target is five, then applying $K=5$ K-means clustering to the bi-clustered attention map may produce poorly separated or degenerate regions.

* How robust is the region planning when the initial attention map is significantly inconsistent with the target count?

* What is the failure rate of K-means clustering in this case? * Did the authors consider iterative refinement of the region planning?

W3. Insufficient Scalability and Computational Cost Analysis

The method includes: (1) an initial $S$-step denoising process for region planning; (2) noise reconstruction and saliency enhancement; (3) gradient-based optimization of the attention map at each step during early denoising. The total computational cost is not reported in the paper.

* What is the cost of wall-clock time/actual time consumption compared to standard generation?

* How does the computational cost scale with the number of target objects $K$?

* How many optimization steps are required per denoising step for the attention-guided consistency strategy?

Given that Counting Guidance (a baseline method) is also gradient-based and has a slower inference speed, runtime analysis is particularly important for a fair comparison.

W4. Excessive Hyperparameters and Lack of Sensitivity Analysis

This method introduces a large number of hyperparameters: $tau$ (attention threshold), $S$ (initial denoising steps), $gamma$ and $beta$ (saliency enhancement factors), $lambda_o$ (Otsu scaling), $lambda_s$ (temperature), $lambda_d$ (positive/negative balance), $eta$ (numerical stability), $eta_t$ (optimization step size), $lambda_1, $lambda_2, $lambda_3$ (loss weights), and the number of early attention optimization steps. The paper does not perform sensitivity analysis on most of these parameters.

Please provide sensitivity curves for the most critical hyperparameters (at least $gamma$, $beta$, $tau$, $lambda_1$, $lambda_2$, and $S$). Discuss whether different base models require different hyperparameter settings.

W5. LRR module makes strong assumptions about the noise structure.

Latent Representation Reorganization (LRR) arranges latent features such that high-attention features are placed near the center of the region. This assumes that: (a) individual spatial features in Gaussian noise carry meaningful object-related information before any denoising is performed; and (b) rearranging them preserves the distributional properties of the noise. Both of these assumptions are questionable.

Feature rearrangement within a region alters local spatial correlation and may break the independent and identically distributed (i.i.d.) Gaussian assumption upon which the denoising process relies. While this method is empirically effective, its theoretical basis is weak.

Please analyze whether the modified noise $z_T'$ still approximately follows a Gaussian distribution (e.g., by the Kolmogorov-Smirnov test) and discuss potential distribution shift issues.

W6. Limited Analysis of Failure Cases

Section 4.5 mentions three limitations (scalability in high-density scenes, sensitivity to complex relational cue words, and quality tradeoffs), but details are postponed to the appendix. Given that the core mechanism of this method is region partitioning, failure modes for large numbers (e.g., >10) or spatially constrained compositions (e.g., “five birds perched on a wire”) deserve explicit discussion.

W7. Incomplete Comparison with Layout-Based Methods

The paper positions itself as a comparison with layout-based methods, but does not compare it with several related methods:

* LayoutGPT + ControlNet/GLIGEN pipelines (which provide explicit bounding box control).

* SLD (Self-Correcting LLM Controlled Diffusion) and similar iterative refinement methods.

Comparisons with Attention Refocus and CountGen are valuable, but these represent only a small fraction of layout-based methods. A more comprehensive comparison would strengthen its claim of achieving state-of-the-art (SOTA) performance.

W8. The proposed STA metric lacks sufficient justification.

The Saturated Truncated Accuracy (STA) metric truncates the relative bias to 100%. While this avoids oversensitivity to extreme outliers, it also obscures meaningful information about a large number of count failures. The paper does not compare STA to existing metrics in the counting domain, nor does it discuss why standard metrics (such as MAE and Accuracy) are insufficient.

---

> ### Author Rebuttal · Authors · 2026-03-31
>
> We appreciate all the constructive feedback from the reviewer.
>
> **Q1: Impact of Initial Attention-Target Alignment.**
>
> **A:** We acknowledge that our method did not explicitly account for such cases. Through extensive qualitative visualization, we observed that while our method remains robust in most such scenarios, redundant interventions can occasionally introduce unnecessary perturbations that lead to counting errors. This suggests that **intervention is not always beneficial when the initial layout is already favorable**.
>
> Motivated by this finding, we refine our framework by introducing a counting-based gate: if the base model already satisfies the target count, we directly use the base model’s output. This adaptive strategy prevents over-correction in successful cases while maintaining full intervention when needed, leading to improved performance across all benchmarks. The updated results are shown below and will be included in the revised version.
>
> **Table 1: quantitative results.**
>
> | Mode | COCOCount (STA / Acc / IR) | T2I-Count (STA / Acc / IR) | MultiCount (STA / Acc / IR) |
> | :--- | :---: | :---: | :---: |
> | **Ours (SDXL)** | 0.89 / 0.7 / 1.07 | 0.85 / 0.64 / 0.74 | 0.72/ 0.29/1.37|
> | **Ours (SD3)** | **0.93** / **0.74** / **1.28** | **0.9** / **0.7** / **0.93** | **0.81** / **0.41** / **1.5** |
>
> **Q2 & Q4 & W2: K-means Clustering and Region Planning**
>
> **A:** Please refer to our response to Reviewer yy9U’s second question.
>
> **Q3: Generalization of the Noise-Guided Layout Prior Across Architectures.**
>
> **A:** Our observations of the noise-guided layout prior extend beyond UNet-based SDXL (Fig. 1) to DiT-based architectures including SD v3 (refer to https://anonymous.4open.science/r/1-E1AE/1.png). To further validate this, we conducted a quantitative analysis on SD v3 following the protocol in Appendix C.2. The results mirror the trends seen in Figure 8 (refer to https://anonymous.4open.science/r/3-8F4D/3.png), demonstrating that the layout prior is a consistent phenomenon across diverse model architectures.
>
>
> **Q5: Generalized to Uncountable Compositional Properties.**
>
> **A:** (1) Our method can be extended to handle attributes (e.g., color), for instance in prompts like “three apples and four pears”, by introducing an additional loss in the second stage to enforce attribute–instance alignment, similar to existing training-free approaches. (2) In contrast, extending to spatial relationships is more challenging, as our method depends on the base T2I model’s priors, which are limited for complex relations. While simple cases (e.g., “three apples on a table”) are supported given a reasonable initial layout, multi-category scenarios with complex relations (e.g., inside) remain challenging due to the partition-based strategy, as discussed in Appendix F.
>
>
> **W1: Overstated Formalization of Noise-Induced Layout Prior.**
>
> **A:** Please refer to our response to Reviewer jehR’s third question.
>
> **W3: Computational Overhead During Inference.**
>
> **A:** Please refer to our response to Reviewer mdPw’s first question.
>
> **W4: Excessive Hyperparameters and Limited Sensitivity Analysis.**
>
> **A:** Please refer to our response to Reviewer jehR’s second question.
>
> **W5: LRR Module Analysis.**
>
> **A:** To address concerns about distribution shift after LRR, we analyze the modified latent by testing (1) Gaussianity (via one-sample K-S, Jarque–Bera, and Anderson–Darling tests) and (2) changes in correlations between neighboring spatial locations. The results show no evident distribution shift: the noise remains close to Gaussian (K-S p=0.76, 100% pass; JB p=0.50, 94.3% pass; A–D rejection 4.1% at 5% level), and local correlations stay near zero before and after LRR. Overall, LRR improves layout formation while preserving Gaussianity and local spatial statistics.
>
> **W6: Limited Analysis of Failure Cases.**
>
> **A:** (1) **High-Count Scenarios**: Failures mainly stem from the limited $64 \times 64$ cross-attention resolution, which hinders precise partitioning of densely packed instances. (2) **Relational Constraints**: For prompts with complex spatial relations, the method prioritizes counting accuracy but does not explicitly model structured inter-object arrangements.
>
> **W7: Incomplete Comparison with Layout-Based Methods.**
>
> **A:** We evaluated our method against the suggested layout-based baselines. Table 2 shows that our approach consistently yields superior performance across all benchmarks.
>
> **Table 2: quantitative results.**
> | Model | COCOCount (STA / Acc / IR) | T2I-Count (STA / Acc / IR) | MultiCount (STA / Acc / IR) |
> | :--- | :---: | :---: | :---: |
> | **LayoutGpt** | 0.71 / 0.52 / 0.02 | 0.66 / 0.48 / -0.23 | 0.46 / 0.11 / 0.05 |
> | **SLD** | 0.71 / 0.41 / 0.52 | 0.64 / 0.33 / 0.08 | 0.50 / 0.10 / 0.97 |
>
> **W8: Lack of Validation for the STA Metric.**
>
> **A:** Please refer to our response to Reviewer jehR’s fourth question.
>
> **Note that all quantitative results are rounded to two decimal places for brevity.**

---

> > ### Author Rebuttal · Reviewer_UAHz · 2026-04-03
> >
> > Most of the questions are solved. I really don't have time to verify its correctness, and I believe the author is honest. I will maintain the positive score. Please promise that you will add all of the extra results tables to your camera-ready version!

---

> > > ### Author Response · Authors · 2026-04-03
> > >
> > > Thank you very much for your positive feedback. We are truly encouraged to hear that our rebuttal has addressed your concerns. We confirm that all additional result tables are accurate and will be included in the camera-ready version. In addition, we will release our complete codebase and configuration files upon acceptance.

---

### Official Review · Reviewer_mdPw · 2026-03-12

**Soundness:** 3
**Presentation:** 4
**Significance:** 3
**Originality:** 3
**Overall Recommendation:** 4
**Confidence:** 4

**Summary:**

This paper investigates the persistent challenge that text-to-image (T2I) diffusion models face in accurately generating a specified number of objects. The authors observe that the initial latent noise exerts a decisive influence on spatial layout formation, formalizing this phenomenon as the "Noise-Induced Layout Prior". Leveraging this insight, the paper proposes a novel training-free framework consisting of two core modules: 1) a Count-Aware Noise Adjustment Strategy, which manipulates initial noise to align layout formation with the target count; and 2) an Attention-Guided Layout Consistency Strategy, which performs test-time optimization on early-stage cross-attention. Experiments demonstrate that this approach consistently outperforms strong baselines in both single-category and multi-category tasks on models such as SDXL and SD v3. Furthermore, these quantitative improvements are corroborated by human visual assessments detailed in the appended User Study.

**Compliance With Llm Reviewing Policy:**

Affirmed.

**Key Questions For Authors:**

1. **Computational Overhead During Inference:** The proposed method requires K-means clustering, Voronoi diagram generation, feature reorganization, and bipartite graph matching using the Hungarian algorithm at each step for multi-category scenarios during the early denoising steps. The paper does not detail how much these operations increase the generation time (latency/throughput) and memory consumption per image. For a training-free framework, inference cost is a critical metric. Could you provide specific overhead comparison data in the rebuttal?
2. **Stability of Region Planning for Amorphous or Boundary-Blurred Objects:** Currently, the 26 categories chosen in the MultiCount Benchmark (e.g., apples, cats, bottles) are mostly rigid or semi-rigid objects with clear physical boundaries. If the model encounters amorphous objects without fixed shapes and blurred boundaries, such as "three clouds" or "two puddles of water," the high-response regions based on cross-attention might be highly dispersed. In such cases, would the current hard-partitioning method based on K-means and Voronoi face significant challenges?
3. **Further Clarification on the "Cartoonish/Grid-like" Limitation:** In Section 4.5 and Appendix F, you point out that the method sometimes results in a "cartoon-like appearance" or a "rigid, grid-like layout". This is a very interesting phenomenon. Is this trade-off primarily related to the Attention Concentration Loss ($\mathcal{L}_{acl}$) forcibly contracting features? Did you observe in your experiments whether moderately relaxing the weight of this loss could alleviate this rigidity to some extent and help recover the photorealistic priors of the base models?

**Limitations:**

Yes.

**Strengths And Weaknesses:**

**Strengths:**
1. **Soundness & Originality:** Attributing the failure of object counting to the "Noise-Induced Layout Prior" is a profound and highly insightful observation. The designed training-free intervention mechanism, which operates directly in the latent space to align object layouts, has clear physical intuition.
2. **Significance:** The paper provides rigorous quantitative analyses on both single and multi-category benchmarks and supplements them with careful user studies (Study I & II). The results indicate that the method significantly improves counting accuracy while effectively maintaining the visual quality of the base models.
3. **Presentation:** The paper is well-structured, and the visual explanations (such as the evolution of cross-attention maps across timesteps) intuitively and strongly support the core arguments.

**Weaknesses:**
1. **Lack of Discussion on Computational Overhead:** Despite being a training-free method, it introduces K-means clustering, Voronoi diagram partitioning, feature reorganization, and the Hungarian matching algorithm during the early denoising stages. The paper lacks a quantitative analysis regarding the resulting inference latency and the additional memory footprint.
2. **Insufficient Exploration of Region Planning Limitations:** Most of the 26 categories selected in the MultiCount benchmark are rigid or semi-rigid objects. The robustness of the current Voronoi-based hard partitioning approach remains under-verified when dealing with amorphous objects that lack clear boundaries.
3. **Compromise on Visual Priors:** The authors honestly point out that the method may lead to "grid-like" or "cartoon-like" appearances, but there is a lack of deeper analysis into the loss function mechanisms driving this trade-off.

---

> ### Author Rebuttal · Authors · 2026-03-31
>
> **Q1: Computational Overhead During Inference.**
>
> **A1:** We thank the reviewer for raising this important question. We randomly sample 90 and 120 prompts with varying counts from COCOCount and MultiCount to evaluate different methods in single- and multi-category settings, respectively. All evaluations are performed on an A800 GPU, and Table 1 reports the per-image generation time and memory usage.
>
> **Table 1: time and memory consumption. S-Time and M-Time correspond to the single-category and multi-category settings, respectively.**
>
> | Model             | S-Time(s) | M-Time(s) | GPU(G) |
> |-------------------|-----------|-----------|--------|
> | SDXL              | 8         | 8         | 16     |
> | SD3               | 10        | 10        | 18     |
> | FLUX              | 14        | 14        | 33     |
> | PixArt-$\Sigma$          | 2.9       | 2.9       | 15     |
> | PlayGround        | 3         | 3         | 11     |
> | LayoutGpt (new)        | 10.6      | 10.8      | 14     |
> | **MaskUnet** (new)         | 61.9      | 61.9      | 18     |
> | **SLD**  (new)             | 49.8      | 64.1      | 28     |
> | **Counting Guidance**  | 13.4      | 15.7      | 15     |
> | **IoCo**              | 117       |    -       | 26     |
> | **CountGen**          | 42        |    -       | 50     |
> | **Ours (SDXL)**       | 36        | 85.6      | 23     |
> | **Ours (SD3)**       | 46        | 110.1     | 50     |
>
> Overall, all training-free methods (highlighted in bold) incur higher time and memory costs than conventional T2I models. In the single-category scenario, our method still has a time advantage over other training-free methods. However, in the multi-category scenario, we assign one latent per category and perform loss guidance sequentially within the same timestep, which roughly doubles the runtime. In the future, this could be improved through parallel computation.
>
> Our SDXL-based method adds 7 GB of memory over SDXL, while the SD3-based version adds 32 GB. This is due to the larger attention scale in SD3: with gradient-based attention guidance, SD3 must store significantly larger intermediate states. Unlike U-Net models that mainly retain cross-attention, SD3 preserves global self-attention, resulting in higher memory overhead.
>
> Moreover, in both single- and multi-category settings, the **Count-Aware Noise Adjustment Strategy** accounts for less than 1.5% of the total runtime, while most of the computation is spent on the **Attention-Guided Layout Consistency Strategy**. Therefore, the overhead of K-means clustering, Voronoi diagram generation and feature reorganization is negligible. The computational cost does not show a significant correlation with the target object count $K$.
>
>
> Lastly, the attention-guided layout consistency strategy is applied only during the first four denoising steps, with at most 20 optimization iterations per step, ensuring that the additional overhead introduced by iterative optimization remains manageable.
>
> These discussions will be incorporated into the revised manuscript.
>
> **Q2: Stability of Region Planning for Amorphous or Boundary-Blurred Objects.**
>
> **A2:** We thank the reviewer for this constructive question. MultiCount primarily includes rigid or semi-rigid objects with clear boundaries. To explore the model's limits, we conducted qualitative tests on amorphous object categories. Our method remains effective for "naturally countable" amorphous instances (e.g., clouds, flames, snow patches) where spatial separability is maintained. However, performance degrades for highly diffuse categories (e.g., mist, fog, puddles of water) due to dispersed cross-attention, which challenges our hard-partitioning logic and leads to object merging. This suggests our framework assumes instance separability, and extending it to highly amorphous objects is an important future direction.
>
>
> **Q3: Further Clarification on the "Cartoonish/Grid-like" Limitation.**
>
> **A3:** We thank the reviewer for this insightful observation. Our ablation and sensitivity analyses confirm that "stylized" or "grid-like" artifacts are primarily tied to the **Attention Concentration Loss** ($\mathcal{L}_{acl}$). Specifically, over-constrained guidance (often triggered by a smaller $\lambda_o$ ) can cause attention to over-concentrate within predefined regions, leading to rigid layouts and reduced natural variability. Qualitative assessment across 450 T2I-Count samples shows that such artifacts occur in only ~8.9% (40/450) of cases, suggesting the issue is relatively localized. This phenomenon aligns with prior findings (e.g., [1]) where aggressive spatial constraints can compromise photorealism. In practice, this reflects a controllable trade-off: moderately increasing $\lambda_o$ effectively alleviates rigidity and restores visual realism while maintaining structural accuracy.
>
> [1] Qiu W, Wang J, Tang M. Self-cross diffusion guidance for text-to-image synthesis of similar subjects[C]//CVPR 2025: 23528-23538.

---

> > ### Author Rebuttal · Reviewer_mdPw · 2026-04-03
> >
> > The rebuttal addresses most of my concerns, I will keep my positive score.

---

> > > ### Author Response · Authors · 2026-04-03
> > >
> > > We sincerely appreciate your positive feedback! Your recognition that our rebuttals addressed your concerns means a great deal to us. We're committed to refining our work further and look forward to continued engagement in advancing this research.

---

### Official Review · Reviewer_jehR · 2026-03-13

**Soundness:** 3
**Presentation:** 3
**Significance:** 3
**Originality:** 3
**Overall Recommendation:** 4
**Confidence:** 4

**Summary:**

This paper addresses the challenge of accurately generating a specified number of objects in T2I diffusion models. Experiments on COCOCount, T2I-Count, and a new MultiCount benchmark using SDXL and SD v3 demonstrate improvements over baselines.

**Compliance With Llm Reviewing Policy:**

Affirmed.

**Final Justification:**

Most of my concerns have addressed. I decide to raise my score.

**Key Questions For Authors:**

Please refer to the weakness part.

**Strengths And Weaknesses:**

**Strengths**
1. The attention guidance losses are well-designed: Attention Concentration Loss focuses attention around region centroids, Dispersion Regularization penalizes spatially over-extended attention, and Semantic Suppression Loss addresses multi-category leakage.
2. The method requires no fine-tuning, no external counting models, and no LLMs during inference. This differentiates it from CountGen (requires training a ReLayout U-Net) and IoCo (requires per-token optimization). The approach operates directly on latent noise and early-stage attention maps.
3. Results are compelling across three benchmarks. Improvements are consistent across both SDXL and SD v3 base models and across single-category and multi-category settings.

**Weaknesses**
1. CountCluster[1] has a very similar idea to the proposed method.  The authors should compare their method with CountCluster, both qualitatively and quantitatively.

2. The method introduces numerous hyperparameters: 10+ hyperparameters (τ, S', γ, β, λo, λs, λr, λd, λ1, λ2, λ3, ηt, etc.). While the ablation study demonstrates the usefulness of the loss components, it does not address the sensitivity of these individual hyperparameters. In practice, this creates a very large search space that is difficult for practitioners to navigate and significantly hampers reproducibility.

3. The paper claims to 'formalize' the Noise-Induced Layout Prior (Sec 3.1) but only provides an informal Observation 1 without formal conditions, proof, or mathematical proposition. This overstates the theoretical contribution.

4. The newly proposed STA metric is used as the primary evaluation metric but receives no validation — no correlation with human judgment is shown, and no analysis demonstrates its advantage over existing metrics like Acc or MAE.

[1]CountCluster: Training-Free Object Quantity Guidance with Cross-Attention Map Clustering for Text-to-Image Generation.

---

> ### Author Rebuttal · Authors · 2026-03-31
>
> We appreciate all the constructive feedback from the reviewer.
>
> **Q1: Missing Comparison with CountCluster.**
>
> **A1:** While CountCluster also uses cross-attention for object counting, our method differs by identifying initial noise as the root cause and addressing it via latent reorganization and attention-guided consistency. Since no official code is available, we ensure a fair comparison by following its benchmark and evaluation protocol. As shown in Table 1, our method consistently achieves **higher accuracy with comparable or better MAE/RMSE**. In our qualitative comparison, we use identical prompts and achieve comparable visual quality with accurate object counts (please refer to https://anonymous.4open.science/r/2-433B/2.png).
>
> **Table 1: quantitative results.**
> | Model        | MAE    | Acc    | RMSE   |
> |--------------|--------|--------|--------|
> | CountCluster (SDXL) | 0.8210 | 0.5575 |1.8490 |
> | **Ours (SDXL)** | 1.2942 | 0.6583 |2.5733|
> | **Ours (SD3)** | **0.5033** | **0.6842** | **1.0368** |
>
>
> **Q2:  Excessive Hyperparameters and Limited Sensitivity Analysis.**
>
> **A2:** While our modular design introduces several hyperparameters, we emphasize that the framework is **robust and easy to tune**. We provide the following clarifications:
>
> (1) **Code Release and Reproducibility.** To ensure full reproducibility, we commit to releasing our complete codebase and configuration files upon acceptance.
>
> (2) **Concentrated Parameter Sensitivity.** Our sensitivity analysis (detailed in Appendix) reveals that performance is primarily governed by only two key parameters: $\tau$ and $\lambda_o$. Most other hyperparameters remain stable across a broad range and can be kept at their default values. This concentration of sensitivity simplifies the tuning process, making the method practical for diverse generation tasks.
>
> (3) **Interpretable Mechanics.** The sensitivity of these parameters aligns with our design. For example, $\tau$ regulates the selection of high-response regions in early-stage attention maps: excessive $\tau$ induces attention sparsity, destabilizing K-means clustering and causing cluster centers to become closer and less distinguishable; insufficient $\tau$ allows low-activation signals to participate in clustering, causing cluster center falls in low-response regions and leading to imbalanced clusters. $\lambda_o$ governs layout consistency strategy: excessive $\lambda_o$ causes over-concentration, resulting in rigid layouts and reduced natural variability, while insufficient $\lambda_o$ weakens regional constraints and may cause multiple instances to be generated within a single region.
>
>
>
> **Q3: Overstated Formalization of Noise-Induced Layout Prior.**
>
> **A3:** We agree that the term “formalize” overstates the nature of our contribution in this context. Our primary objective was to empirically characterize and identify the Noise-Induced Layout Prior phenomenon rather than to provide a formal mathematical theorem. We will revise the manuscript as follows:
>
> (1) Replace “formalize” with “characterize” in Sections 3.1.
>
> (2) Retitle 'Observation 1' to explicitly indicate empirical grounding (e.g., “Empirical Observation 1”).
>
> (3) Add a paragraph clarifying that this observation is supported by: (a) controlled visual experiments (Figs 1, 7)；(b) quantitative KL divergence analysis (Fig.8, Appendix C.2), and (c) extensive ablation studies (Sec. 4.4, Table 4)
>
> We believe these revisions better reflect our insight: empirically identifying a key, underexplored factor in counting errors that directly motivates our training-free solution. The method’s utility and novelty remain unchanged.
>
>
> **Q4:  Lack of Validation for the STA Metric.**
>
> **A4:** To validate STA, we conducted a human study on 120 generated images across target counts ${2, 3, 4, 5, 8, 10}$, each evaluated by 20 annotators. As shown in Table 2, both STA and F1-score exhibit significantly stronger correlations with human judgment than traditional metrics such as Acc. We highlight two key insights regarding STA’s utility:
>
> (1) **Superiority over Hard Metrics**: Standard accuracy (Acc) is overly binary; it rewards only exact matches. In contrast, STA captures degrees of correctness, better aligning with human perception in complex scenes.
>
> (2) **Practical Interpretability**: While F1 shows slightly higher statistical correlation, STA provides a more intuitive and linear measure of relative counting error. For example, generating 9 out of 10 objects yields $\text{STA} = 0.90$. By explicitly modeling this “proportionate success” , STA offers a clearer and more practical measure of controllability than F1.
>
>
> **Table 2: Human judgment.**
> | Metric   | Person | Spearman | Kendall |
> |----------|--------|----------|---------|
> | STA      | 0.78 | 0.77     | 0.70 |
> | F1       | **0.83**   | **0.78**  |**0.71**|
> | Acc      | 0.60    | 0.72     | 0.66|
> | Precison | 0.62   | 0.51     | 0.46    |
> | Recall   | 0.42   | 0.45     | 0.43    |

---

> > ### Author Rebuttal · Reviewer_jehR · 2026-04-08
> >
> > I raised my score to weakly accept.

---

> > > ### Author Response · Authors · 2026-04-08
> > >
> > > We sincerely appreciate your positive feedback! Your recognition that our rebuttals have addressed your concerns means a great deal to us. If you are satisfied with our response, could you please adjust the "overall recommendation" score in the system? Very Thanks : )

---

### Decision · Program_Chairs · 2026-04-30

**Decision:**

Accept (regular)

**Comment:**

Initially, this paper received diverse ratings. Reviewer jehR, who was initially negative, was satisfied with the rebuttal, and stated the rating will be upgrated. Reviewer yy9U also upgrated the rating to weak accept.
Reviewer UAHz did not check the correctness of the rebuttal. The AC checked the rebuttal: most feedback is fine. Reviewer mdPw kept the positive score. The AC agreed with the reviewers.